# Multi-thermals and high concentrations of secondary ice: A modelling study of convective clouds during the ICE-D campaign

Zhiqiang Cui[1], Alan Blyth[1,2], Yahui Huang[1], Gary Lloyd[3], Thomas Choularton[3], Keith Bower[3], Paul Field[1,4], Rachel Hawker[1], Lindsay Bennett[2]

[1]Institute for Climate and Atmospheric Science, University of Leeds, Leeds, LS2 9JT, UK
[2]National Centre for Atmospheric Science, Leeds, LS2  9PH, UK
[3]Centre for Atmospheric Science, University of Manchester, Manchester, M13 9PL, UK
[4]Met Office, Exeter, UK

*Correspondence to*: Zhiqiang Cui (z.cui@leeds.ac.uk)

**Abstract.** This paper examines the mechanisms responsible for the production of ice in convective clouds influenced by mineral dust. Observations were made in the Ice in Clouds Experiment – Dust (ICE-D) field campaign which took place in the vicinity of Cape Verde during August 2015.  Measurements made with instruments on the FAAM aircraft through the clouds on 21 August showed that ice particles were observed in high concentrations at temperatures greater than about -8 °C.
Sensitivity studies were performed using existing parameterization schemes in a cloud model to explore the impact of the freezing onset temperature, the efficiency of freezing, mineral dust as efficient ice nuclei, and multi-thermals on secondary ice production by the rime-splintering process.

The simulation with the default Morrison microphysics scheme (Morrison et al., 2005) that involved a single thermal produced a concentration of secondary ice that was much lower than the observed value of total ice number concentration.
Relaxing the onset temperature to a higher value, enhancing the freezing efficiency, or combinations of these, increased the secondary ice particle concentration, but not by a sufficient amount.  Simulations that involved only dust particles as ice nucleating particles produced a lower concentration of secondary ice particles, since the freezing onset temperature is low. The simulations implicate that a higher concentration of ice-nucleating particles with a higher freezing onset temperature
may explain some of the observed high concentrations of secondary ice. However, a simulation with two thermals that used the original Morrison scheme without enhancement of the freezing efficiency or relaxation of the onset temperature produced the greatest concentration of secondary ice particles. It did so because of the increased time that graupel particles were exposed to significant cloud liquid water in the Hallett-Mossop temperature zone.  The forward-facing camera and measurements of the vertical wind in repeated passes of the same cloud suggested that these tropical clouds contained
multiple thermals. It is possible of course that several mechanisms, some of them only recently discovered, may be responsible for producing the ice particles in clouds. This study highlights the fact that the dynamics of the clouds likely play an important role in producing high concentrations of secondary ice particles in clouds.

## 1. Introduction

Ice particles in clouds contribute to the formation of more than half of the world's precipitation and greatly enhance the amount of precipitation process over the warm rain only process (McFarquhar et al., 2017). Precipitation in deep convective clouds is closely related to the riming and melting of ice particles (Cui et al., 2011). Ice phase processes in clouds affect not only the weather but also the climate, which is a new frontier of research in terms of aerosol-cloud-climate interactions (Seinfeld et al., 2016; Storelvmo et al., 2017).

Cloud drops form on cloud condensation nuclei (CCN), and primary ice particles on ice nucleating particles (INPs). The main INP types include mineral and desert dust, metals and metal oxides, organics and glassy particles, bioaerosol, soil dust, biomass and fossil fuel combustion aerosol particles, volcanic ash particles, and crystalline salts (Kanji et al., 2017). The onset and median freezing temperatures of those INPs show great intra- and inter-type variabilities (Kanji et al., 2017). An upper limit of the starting freezing temperatures of some bacteria INPs is typically between $-2$ to $-4$ ºC (Szyrmer and Zawadzki, 1997; Morris et al., 2008), and some even as high as $-1.5$ ºC (Kim et al., 1987, Lindow et al., 1989). The activation temperature of mineral dust is less than $-15$ ºC (Hoose and Möhler, 2012), but depends on their mineralogy (Murray et al., 2012). However, dust particles can serve as INPs at higher temperatures if the fraction of potassium-rich feldspar (K-feldspar) is high because feldspar particles have high ice-nucleating active sites (Atkinson et al., 2013). A full functionalization of the ice-nucleating sites of a feldspar particle with hydroxyl groups enables a strong bonding to the prismatic plane of ice and prompts the formation of ice crystals on the surface of the feldspar (Kiselev et al. 2017), which favours a higher freezing temperature. A recent study showed that a microcline mineral (a potassium-rich alkali feldspar) has bulk freezing temperatures even greater than $-3$ ºC (Kaufmann et al., 2016). It has also been found that sea-spray aerosol particles can serve as INPs (e.g., Wilson et al., 2015, DeMott et al., 2016), but they do not act until the temperatures fall below -10 ºC. Burrows et al. (2013) suggested strong regional differences in the importance of marine biogenic INP relative to dust INP.

Nickovic et al. (2012) developed a high-resolution global dataset of mineral composition and showed the effective mineral content in soil for quartz, illite, kaolinite, smectite, feldspar, calcite, hematite, gypsum and phosphorus. For example, the gradients of the surface content of feldspar are particularly strong in Central Sahara. What makes the INPs even more complex is that they can be chemically and physically modified through chemical processing, internal and/or external mixing, and cloud processing (Kanji et al., 2017).

Ice particles form via various primary pathways with the help of INPs, such as deposition ice nucleation, contact freezing, immersion freezing, condensation freezing, collisional contact freezing, and inside-out evaporation freezing (e.g., Cooper, 1986). Ice particles also form as a results of homogeneous freezing at temperatures about -40 ºC. The INP concentrations are

highly variable at a particular temperature that depends on the dynamics and the aerosol properties and humidity conditions. Most, if not all, models represent the widespread relationships with parameterization schemes. There is no one-size-fits-all solution to parameterize freezing processes for all INPs.

Apart from the primary freezing processes, new ice crystals can be produced by secondary ice production (SIP) in the presence of preexisting ice without INPs. There are several SIP processes: fragments emitted from freezing large drops (e.g., Wildeman et al., 2017), the mechanical breakup of ice crystals by collision with other particles (e.g., Knight, 2012), splinter formation during the riming process (Hallett and Mossop, 1974), enhanced ice nucleation in regions of spuriously high supersaturations (Hobbs and Rangno, 1990), ice particle fracture during evaporation of ice particles (Oraltay and Hallett, 1989; Bacon et al., 1998). Of those processes, the Hallett-Mossop (HM) parameterization has been studied the most and is routinely incorporated in models. New parameterizations for other mechanisms have been developed recently (e.g., Phillips et al., 2017; Lawson et al., 2017). Lawson et al (2015) developed a parameterization for the drop-freezing secondary ice production process and subsequent riming. Lawson et al. (2017) developed an expression that predicts the level in the updraft core, where liquid water becomes depleted. Observational and modelling studies have shown that the observed high ice concentration in some clouds may be explained with the Hallett-Mossop process (e.g., Huang et al., 2008; Crosier et al., 2011; Crawford et al., 2012; Huang et al., 2017; Hawker et al., 2021). For a complete review of the literature on secondary ice production, including the current state and recommendations for future research, see Field et al. (2017), Korolev and Leisner (2020), and Korolev et al. (2020).

Many numerical models, including some cloud models, do not explicitly include the information about CCN and INP, but rather use parameterizations to represent a "general" or a best-fit case for a freezing mode in the primary ice production. As a result, a parameterized microphysics scheme does not reflect the aerosol environment of a cloud. Any departure of the INP properties from the best-fit condition, such as the onset freezing temperature and the freezing efficiency, may lead to an unrealistic representation of the cloud microphysics.

Ice production in clouds is affected not only by microphysical processes, but also by cloud dynamics. Previous studies have revealed that thermals are the building blocks of convective clouds (Koenig, 1963; Blyth et al., 1988; Keller and Sax, 1981; Blyth et al., 2005; Damiani et al., 2006). The clouds often contained multiple thermals that ascended in the wake of their predecessors. In a theoretical study using a detailed microphysics model in a simple dynamical framework, Blyth and Latham (1997) found that multi-thermals can help yield rapid ice HM multiplication by providing a new source of liquid water content and by allowing the particles to be carried by the thermal circulation. The results were consistent with those found earlier by Koenig (1963) and others. The operation of the HM process needs the coexistence of graupel, large drops (> 25 µm) and small drops (< 12 µm) in the temperature zone of -3 °C – -8 °C. Previous studies have investigated the conditions favourable for secondary ice production: moderate vertical velocities to allow graupel particles fall into the HM zone, availability of both large and small drops for riming (e.g., Huang et al., 2008; Heymsfield and Willis, 2014; Huang et

al., 2017). It is important to consider the possibility that multiple thermals will ascend through the HM zone because of the additional source of cloud drops that can rime onto graupel particles.

The field campaign of ICE-D (Ice in Clouds Experiment – Dust ) took place in the Cape Verde region, downwind of the African dust sources, in August 2015. Although there have been previous field campaigns that have investigated ice nucleation in and near the Saharan region, most of them focused on chemical and physical properties of the dust particles and their impact on large-scale phenomena (e.g., Knippertz et al., 2011; Rocha-Lima et al., 2018). The ICE-D was specifically designed to study how dust affected ice production in convective clouds using aircraft measurements, radar, and ground-based aerosol instruments, which was complementary to a series of projects on layer clouds (ICE-L, see Heymsfield et al., 2011) and tropical towering clouds (ICE-T, see Heymsfield and Willis, 2014).

Aircraft measurements were made by the Facility for Airborne Atmospheric Measurements (FAAM) BAe 146 research aircraft on 21 August 2015 of convective clouds about 150 km from the Praia airport, Cape Verde. There are two main points from the observations. Firstly, the maximum concentration of ice particles was greater than 200 per litre, greater than expected from INP alone. Secondly, the first ice particles were believed to have formed at a temperature greater than about -5 °C (Lloyd et al, 2020), which was thought to be a result of biological material either on or internally mixed with the dust particles. This would mean that the ice embryos would form at only slightly supercooled temperatures (Obata et al., 1999; Lloyd et al, 2020)..

In this paper, we report on a modelling study designed to explore several aspects of the production of ice particles. Namely: the impact of the freezing onset temperature; the impact of the efficiency of freezing in microphysics schemes; to inspect the relationship between the measured dust as efficient INPs and the secondary ice production; and to investigate the role of multi-thermals in producing high secondary ice concentrations. A three-dimensional cloud model was used for the simulations. Section 2 describes the observations and datasets. The model description and numerical designs are given in Section 3.  Section 4 presents the simulation results. The final section provides the summary and conclusions.

**2. Summary of the observations**

The observations of the ICE-D field campaign involved an aircraft, radar and ground-based measurements. The UK's FAAM BAe-146 research aircraft was used to conduct airborne measurements of cloud microphysics and aerosol. A suite of instruments onboard the BAe 146 measured the information about cloud microphysics, aerosol particles, and other atmospheric variables (see Price et al., 2018, Liu et al., 2018, and Lloyd et al., 2020 for further descriptions).  One airborne aerosol instrument was the Passive Cavity Aerosol Spectrometer Probe (PCASP), which can measure both aerosol number and size in the nominal size range 0.1 to 3 micrometres (Price et al., 2018; Liu et al., 2018). Ice particle concentrations were

derived from the two-dimensional stereo (2D-S) Probe, an optical imaging instrument that obtains stereo cloud particle images and concentrations using linear array shadowing (Lawson et al., 2006; Cui et al., 2012; Cui et al., 2014). The UK Met office ALS450 lidar manufactured by Leosphere with an emitted wavelength of 354.7 nm and a receiver bandwidth of 0.36 nm was on board the aircraft to measure cloud top height, range corrected signal, relative depolarisation ratio to map cloud and aerosol layers  and to retrieve aerosol optical properties (Marenco et al., 2013). A mobile dual-polarisation Doppler X-band weather radar (Neely III, et al., 2018) of the National Centre of Atmospheric Science, UK was deployed at the Praia airport on the island of Santiago, Cape Verde. A suite of instruments was positioned at the airport to measure aerosol properties (Marsden et al., 2019).

A trough existed along the west coast of Africa (Figure 1a) on 21 August 2015. Convective cloud systems and some smaller and more isolated convective clouds developed along the coast. The aircraft operated in the region from longitude 23.541° W to 21.022° W and from latitude 13.494° N to 15.024° N. The operation region was characterized by relatively low aerosol optical depth (AOD), compared with the dust-plume region between -20° W - -5° W and 18° N – 26° N (Figure 1a). The aerosol subtype image by CALIPSO overpass to the west of the region indicates the existence of a dust layer (in yellow) about 2-km thick (Figure 1b). The aerosol below the dust layer was dominated by clean marine aerosol with some polluted dust. Images from the on-board Lidar show that the range-correlated Lidar signal, which is an indicator of aerosol loading, was much stronger below 1.5 km (all altitudes are above mean sea level (MSL) in this paper) just after taking off. However, there was a layer with a slightly weaker signal at about 2.3 km during the profile in Figure 1c, which was approximately 150 km to the west of the clouds shown in Figure 1a. The relative depolarization ratio (spheres close to 0 and non-spheres much higher) indicates the aerosol particles were soluble below 1.5 km within the boundary layer and insoluble in the aloft layer (Figure 1d). The Lidar signals were consistent with the PCASP measurement in Fig. 1e, which also shows the two aerosol layers. Together with the CALIPSO image, we can conclude the aloft aerosol layer was mostly dust.

The aircraft penetrated clouds between 100 and 300 km to the southeast of Praia at various levels in the ambient temperature range of -1 to -7 ºC. In-cloud temperatures could not be obtained because of wetting problems.  The aircraft followed the ascending cloud top wherever possible and made the passes a few hundred metres below the top in order to detect the formation and production of primary and secondary ice particles.

Figure 2 shows the concentrations of ice particles in several passes based on the aircraft measurements. The maximum concentrations of ice particles (i.e., non-spherical in shape) in the size range of 50 -1280 μm were 44 $L^{-1}$ for a pass at -3.1 ºC (~5500 m), 52 $L^{-1}$ at -4.4 ºC (~5800 m), 270 $L^{-1}$ at -4.7 ºC (~6100 m), and 82 $L^{-1}$ at -6.8 ºC (~6500 m). As an example, the time series of concentration and vertical velocity for the pass with the highest concentration is shown in Figure 3. The observation indicates that the ice concentrations were a few tens per litre at derived temperatures in cloud between 0 and −2

165 °C (Lloyd et al., 2020). It is impossible of course to be certain of the origin of ice particles in such clouds. The fact that the passes were made within a few hundred metres of cloud top, the concentrations of ice particles were higher than expected from typical INP measurements for the estimated cloud-top temperatures, and there was no evidence of higher concentrations of ice particles in the downdraughts suggest that secondary ice production most likely occurred.

## 3. Cloud Modelling

### 3.1 Model description

The Cloud Model 1 (CM1) was used for simulations in this study. More details on the model can be found in Bryan et al. (2003) and Bryan and Morrison (2012). The model uses conserved mass and energy conservation numerical schemes in 3-dimensions and has a rich choice of microphysics schemes. In our simulations, the Morrison double-moment microphysics scheme (Morrison et al., 2005) was used to predict the mass ratio and the number concentration of cloud droplet, rain, ice,

snow and graupel. The microphysical processes of drops include condensation, evaporation, collision and coalescence, sedimentation of cloud particles, particle growth by deposition of water vapour. The processes involved with ice include primary freezing modes of deposition/condensation, contact, and immersion, and secondary freezing through the riming-splintering (the Hallett-Mossop) process (Hallett and Mossop, 1974; Cotton et al., 1986), in the temperature range of -3 to -8 °C with a maximum at -5 °C, and the transition and interaction between different species.

### 3.2 Experimental design

One of the objectives of our study is to investigate the impact of the onset freezing temperature and the freezing efficiency on secondary ice production. As mentioned in the introduction, the onset temperatures vary with the type of the INPs. The Morrison scheme uses the Cooper parameterization (Cooper, 1986), in which freezing begins at a temperature of -8° C. The temperature where freezing begins depends on the INP types. Recently, Garimella et al. (2018) summarised some of the

185 limitations of field measurements of INP and its influence on the simulated cloud forcing in a global model. The parameterizations derived from field and laboratory data using the continuous flow diffusion chambers (CFDCs) for example, are subject to systematic low biases due to the limit of detection, not experiencing the maximum supersaturation, and the concurrence of ice particles with drops. To reduce the bias, DeMott et al. (2015) proposed to apply a calibration factor of 3 to multiply the measured INP to obtain a better agreement. However, Garimella et al. (2017) noted that the

190 calibration varied from 1.5 to 9.5 because of the lower relative humidity with respect to water than the intended values if aerosol deviated from the laminar flow, which indicates that this is one of the major problems in quantifying the formation of ice in numerical models.

Considering these uncertainties, we investigate the influence of the onset temperature and the efficiency on the ice

production in sensitivity simulations using the Morrison scheme as the *control* run, where the initial drop number

concentration is 150 cm$^{-3}$. We also modify the Cooper parameterization in the Morrison scheme with a parameterization of DeMott et al. (2010) based on several datasets from different regions as a function of temperature. A recently-developed parameterization by Paukert and Hoose (2014) is also tested to probe whether dust INPs alone can produce the concentration of ice observed by the aircraft. The details of the *control* and the sensitivity tests are given in Table 1. The Morrison scheme has several ice freezing modes, including immersion freezing, deposition freezing as a function of supersaturation with respect to water and ice for this scheme, contact freezing, homogeneous freezing, and the secondary ice production by the HM process. For relaxation and enhancement sensitivity simulations, we only modified the immersion freezing mode. The aims of the sensitivity simulations are summarised as follows. The *early onset1* run examined the effect of active INPs at higher temperatures on secondary ice production when the onset temperature was increased to -3 °C. The *Cooper10x* run explored the effect of more INPs (i.e., the freezing efficiency was multiplied by 10). The *early ohnset1 & Cooper10x* run combined effects of the above two, while the *early onset1* & *100xINP* run and the *early onset2* & *100xINP* run probed the effect of even higher loadings of INPs. The DeMott scheme (2010) was examined in runs *Demott*, *early Demott*, and *Demott 10xINP*. To investigate the effect of the dust as INP, the Bigg (1953) scheme was replaced by the Paukert and Hoose scheme (2014) since the Bigg scheme does not consider INP types, but the Paukert and Hoose scheme considers different INP types. The *Paukert* run used the mineral dust parameters in the Paukert and Hoose scheme. The *Paukert-dust* run was same as the *Paukert* run except that the INP numbers were increase by a factor 3.3 in the layer between 2–3 km where the dust layer the coarse mode concentrations in that dust layer were approximately 3.3 times higher than those above the layer. (Figure 1e). Finally, the effect of multi-thermals on secondary ice production was examined in *multi-thermals* when a second bubble of 2 °C was added after 20 min into the simulation.

## 3.3 Model setup

The domain contained 100×100 grid points in the horizontal direction and 80 levels in the vertical direction, with a grid spacing of 150 m in the three directions. The time-step was 2 sec, and the output frequency was 1 minute. Most of the simulations had a duration of 60 min except for the multi-thermal and the Paukert-scheme runs which had a 120 min duration. The model was initialised with a horizontally homogeneous atmosphere (Figure 4). Initial profiles of potential temperature and water vapour mixing ratio were taken from measurements made by radiosondes released from the aircraft in the vicinity of the clouds studied. Because the release level of the dropsonde is lower than the highest level of the model domain, we used the NCEP/NCAR reanalysis data close to the cloud to represent the air above the radiosonde drop-off level. The simulated clouds were triggered by a warm bubble of 2 ºC in the *control* and most of the sensitivity runs except the multi-thermal run where another bubble was added 20 min from the starting time.

## 4. Results

### 4.1 Control simulation

Figure 5 shows a time sequence of a cross-section along the centre of the simulated cloud for the control run (*control*) with only one thermal. The cloud ascended generally at a rate of 300 m per minute between 20 and 25 min. The maximum vertical wind was 17.7 ms$^{-1}$ at 20 min and 8.3 ms$^{-1}$ at 25 min; thereafter, it ascended at a rate of about 150 m per minute to 36 min. The maximum vertical wind was 6.4 ms$^{-1}$ at 30 min and 4.4 ms$^{-1}$ at 35 min. The maximum level of cloud top was 9075 m at 36 min. Figure 5a shows that there was a column of supercooled raindrops up to a temperature of about -5 °C (6.3 km). There were no ice crystals or graupel particles. The cloud top temperature was about -6 °C at the time. The cloud top reached around 8 km (T ~ -14 ºC) at 25 minutes. Ice particles were present in the upper 500 m with a maximum concentration of 0.5 L$^{-1}$, but there were no graupel particles. The column of supercooled raindrops reached a temperature of about -14 °C. There were few, if any observations of such cold columns, especially in tropical oceanic clouds. At 30 min, the graupel particles fell into the HM zone, presumably around the edges of the thermal in the downdraughts and then at the rear of the thermal where the updraught is much weaker. The arrival of the graupel in the HM temperature zone allowed for splinters to be produced by the HM process. The graupel and ice concentrations in the HM zone reached 0.35 L$^{-1}$ and 4.1 L$^{-1}$, respectively. The cloud had further developed by 35 min such that the top had reached an altitude of about 9 km (T ~ -22 ºC); the maximum height. The concentrations of ice in the HM zone and at the cloud top were 2.1 L$^{-1}$ and 14.5 L$^{-1}$, respectively. By this time the entire cloud-top region had begun to descend. The cloud top continued to descend to about 8.5 km (T ~ - 18 ºC) by 40 min. The ice crystal concentration increased to 21.5 L$^{-1}$ at cloud top.

The maximum values at each level in the *control* run of the vertical velocity, and the number concentrations of raindrops, graupel particles and ice crystals are shown in Figure 6. As the cloud developed, latent heat release and reduced water loading drove the increase in the vertical velocity, the maximum reached 18.8 ms$^{-1}$ at z = 5.55 km at 19 min. Raindrops developed after 7 min, and the maximum concentration was 330.4 L$^{-1}$ at z = 3.45 km at 13 min. The major region of graupel particles appeared at an altitude of approximately 8 km at about 30 min, with the maximum being 1.7 L$^{-1}$ at 8.7 km at 41 min. Additionally, a local maximum was at about 6 km and its value was 0.26 L$^{-1}$ at 31 min. The major region of ice crystals occurred above an altitude of 8 km with a maximum of 21.5 L$^{-1}$ at z = 9.15 km at 41 min. In the HM zone, the maximum concentration was 4.1 L$^{-1}$ at z = 6.6 km at 30 min, closely related to the local maximum in graupel concentration. Figures 6e and 6f show the variations of the ice particle concentration of the *early onset1* run and the *early onset1 & noHM* run, respectively. The maximum ice particle concentration in the HM zone of the early onset1 simulation was 8.64 L$^{-1}$, while it was 0.27 L$^{-1}$ of the early onset1 & noHM simulation, which clearly indicates the importance of rime splintering in the HM zone where it is active more directly.

## 4.2 Maximum concentration in the HM zone

Figure 7 shows the temporal variation of the maximum concentration of ice in the HM zone for the *control* and sensitivity runs. Overall, the concentrations started to increase at about 20 min and reached the maximum values at about 30 min except in the *multi-thermals* run. The curve of concentrations for the *multi-thermals* run was identical to that of control run before 45 min but greatly increases afterward.

The *control* run produced a maximum ice concentration of 4.1 $L^{-1}$ which was significantly smaller than the observed value (> 40 $L^{-1}$). The Morrison microphysics scheme uses the Cooper parameterization for the primary ice production, which is based on a best fit curve of measurements from Wyoming wintertime cap clouds, wintertime orographic clouds of southwestern Colorado, Israel winter cumulus clouds, summertime cumulus clouds of Montana, cumulus clouds of South Africa; and Australian cumulus clouds. As discussed by Cooper (1986), the variance in the ice concentrations at any given temperature is 270 large, probably due to the high variability in the INP population itself or the wide variability in the activated fraction of INPs. Since the chemical and physical properties of INPs vary with time and space, the Cooper parameterization does not necessarily represent the INP conditions of the observed cloud. To investigate the impact of possible INP properties, such as the onset freezing temperature, the abundance of active INPs, and the freezing efficiency on the ice concentration in the HM zone, a series of sensitivity simulations w conducted, as plotted in Figure 7. The maximum ice concentration was twice 275 higher when the rate of ice production in the HM scheme was doubled. When the onset freezing temperature of the Cooper parameterization in the Morrison scheme was relaxed from the default value of -8 ℃ to -3 ℃ (*early onset1*), the maximum concentration increased to 8.8 $L^{-1}$. If the ice number concentration produced with the Cooper parameterization was multiplied by 10 (*Cooper10x*), the maximum concentration increased to 13.4 $L^{-1}$. A combination of the relaxation and enhancement (*early onset1* & *Cooper10x*) further increased the concentration in the HM zone to 30.6 $L^{-1}$. The maximum 280 concentrations increase to 45.3 $L^{-1}$ in *early onset1* & *100xINP* and 48.6 $L^{-1}$ in *early onset2* & *100xINP*. The DeMott scheme (*Demott*) produced the maximum concentration of 11.7 $L^{-1}$. It increased to 32.2$L^{-1}$ with relaxation (*early Demott*) and 40.4 $L^{-1}$ with both relaxation and enhancement (*Demott 10xINP*).

The maximum concentration in the HM zone decreased to 1.8 $L^{-1}$ using the Paukert scheme (*Paukert*) when the Bigg (1953) 285 scheme was replaced by the Paukert and Hoose (2014) scheme (*Paukert*). Accounting for the dust layer (*Paukert-dust*) led to only a slight increase to 1.9 $L^{-1}$.

Overall, the greater the starting freezing temperature, the higher the maximum ice concentration appears in the HM zone. Simulations with both relaxation and enhancement can produce total ice concentration observed in some passes (e.g., ~ 50 $L^{-1}$ 290 in pass at approximately 5830 m in Figure 3), but much lower than the maximum value ~ 270 $L^{-1}$ observed in the cloud.

The temporal variations of the ice production in the HM zone were broadly similar except for the two-thermal run (*multi-thermals*) which produced a maximum concentration of 121 $L^{-1}$ at 69 mins. We will discuss the causes of enhanced secondary ice production in the microphysical sensitivity runs in Sections 4.3 – 4.5.

**4.3 Onset freezing temperature and freezing efficiency**

Sensitivity tests were used to investigate the effect of varying the onset freezing temperature and freezing efficiency on secondary ice production. The differences of several microphysical properties between the sensitivity tests and the *control* simulations are examined. There was a significantly higher concentration of secondary ice particles produced in *early onset1* & *Cooper10x* compared to the *control* run. Figure 8 shows the detailed differences between the two runs. A banana-shaped area of positive difference in vertical velocity appeared in Fig. 8a. The positive region increased with time and height from 23 min and z ~ 6.6 km, reaching a maximum concentration of about 2.7 m $s^{-1}$ at 35 min at 9.15 km. The increases in the vertical velocity were most likely caused by the extra latent heat release due to the enhancement of freezing (e.g., McGee and van den Heever, 2014).

The region of the positive difference in the cloud water mixing ratio was well correlated with the enhanced vertical motion before 35 min, indicating that the enhanced vertical velocity pushed the cloud top higher, which was confirmed with an inspection of the fields of the two simulations and their difference (figures now shown).

Although there was a small area of increase in the rain mixing ratio, the ratio generally tended to decrease mainly below 7 km and with the minimum at 24 min and z = 6.6 km. The decrease was a result of more raindrops being converted to graupel particles.

The maximum increase in the raindrop number concentration in *early onset1* & *Cooper10x* wass at 30 min and z = 8.55 km (Figure 8e). However, a decrease occurred after about 37 min and the minimum is above 8 km. The positive and negative regions shown in Fig 8e is explained by the fact that the concentration of raindrops was greater in the onset1 & Cooper10x than in the *control* run up to about 38 min and these drops remained elevated thereafter while those in the *control* run began to fall (figures not shown).

The mixing ratios of graupel and the ice crystals tended to increase (Figure 8c and d, respectively). The maximum increases appeared at 24 min and z = 6.75 km and at 47 min and z = 9.45 km, respectively. The increase in the mixing ratio of graupel seemed to be related to the raindrops being converted to graupel by direct freezing since there was no increase in ice particles at that time and altitude. The graupel number concentration increased (Figure 8f), with the maximum difference being at 39 min and z = 9.15 km.

Differences in the maximum ice crystal concentration at each model level as a function of time between all the sensitivity runs and the control are shown in Figure 9.

For the *early onset1* run, the differences at the upper levels were small since the onset freezing temperature was only relaxed from -8 to -3 °C. However, the increase in the ice crystal number concentration in the HM zone was 5.9 $L^{-1}$. For the
enhancement run *Cooper10x*, the concentration increased both in the upper levels and in the HM zone due to more primary ice production, and the latter is 13.4 $L^{-1}$. With both relaxation and enhancement, the increase in the HM zone was 30.6 $L^{-1}$. The relaxation led to earlier secondary ice production and the enhancement increased ice production not only in the HM zone but also in the higher levels. The differences in concentrations further increased to 45.2 $L^{-1}$ in *early onset1* & *100xINP* and 48.5 $L^{-1}$ in *early onset2* & *100xINP*, as expected. There were more (less) increases in the lower (upper) levels when
using the DeMott scheme (Demott) because of the slope of the DeMott curve, i.e., more IN at higher temperatures.The ice concentration in the HM zone in *Demott* was 11.6 $L^{-1}$. When the onset freezing temperature was relaxed from -8 to -3 °C (*early Demott*), there was a slight increase at the upper levels, but a larger increase in the HM zone (32.1 $L^{-1}$). With both relaxation and enhancement (*Demott 10xINP*), the increase in the HM zone was even larger (40.3 $L^{-1}$), although not as large as in the *early onset1* & *100xINP* or *early onset2* & *100xINP*.

The results indicate that combinations of relaxing the onset freezing temperature and enhancing the freezing efficiency can producesecondary ice in concentrations of several tens per litre. However, these concentrations are significantly less than the maximum concentration observed by the aircraft.

## 4.4 Dust particles as INP

It is interesting to consider if it was possible to reproduce the observed concentrations of ice particles via primary and secondary ice production processes if the INP were only dust particles? We used the Paukert scheme to address this question with two simulations (see Table 1). The differences in the vertical velocity between the *Paukert* and control runs were less than 0.7 $ms^{-1}$ (Figure 10a). The number concentrations of raindrops, graupel particles, and ice crystals generally increased approximately above 8 km and decreased slightly below (Figures 10b-10d). There was less riming, therefore, less secondary
ice production in the HM zone. The decrease in the ice concentration in the HM zone was about 3 $L^{-1}$. This was because that the onset freezing temperature in the Paukert scheme was -12 °C, less than the default value, -8 °C, in the Morrison scheme. Freezing took place later in time and hence higher in altitude.

To account for the dust layer between 2 and 3 km (Figure 1), we increased the INP number to drop number ratio by a factor
of 3.3 in the Paukert scheme. However, the results indicate that there was an insignificant increase in the concentration of ice particles (figures not shown). The results of these two simulations suggest that dust alone as a source of INPs was not enough to produce secondary ice concentrations that were similar to the observations in this case.

## 4.5 Multiple thermals

The above results indicate that the freezing rate and onset temperature in the Morrison scheme affect the secondary ice production. However, none of them produce a sufficient number of ice particles to explain the observations. Next, we will discuss the impact of cloud dynamics on the secondary ice production in a cloud with multi-thermals. It is important to consider both the dynamics and microphysics and their interactions since both play a critical role in ice production. Figure 11 shows the time-height variations of maximum values of the vertical velocity, the raindrop concentration, the graupel particle concentration, and the ice crystal number concentration in the *multi-thermals* run.

The defining feature of this run was the two updraughts (Fig. 11a). The isoline of 2 ms$^{-1}$ of the first updraught started from the beginning at about 1 km and ended at about 35 min and reached an altitude of approximately 9 km, whilst the second started at about 20 min and also reached approximately 9 km. There were no differences in the maximum vertical velocity in the first updraught before 20 min between the *multi-thermals* and control runs. The difference in the first updraught was minimal (< 0.5 ms$^{-1}$) and appears beyond 20 min. Although the maximum vertical velocity in the second updraught was smaller, the updraught lasted for approximately 10 min longer than the first updraught.

There was virtually no difference in the raindrop concentration associated with the first updraught. However, there were two local maxima associated with the second updraught: one between 30 – 40 min at z = 2 – 4 km, and the other between 50 – 60 min at z = 7 – 9 km. In the *control* run, many raindrops precipitated before 20 min (Figure 6b). Some raindrops were transferred to graupel particles via direct freezing, but there were few raindrops remaining near cloud top  (e.g., Figure 5c). In the multiple thermal run, the second thermal started 20 min from the beginning. The lower-level maximum around 30 - 40 min was related to raindrops developed with the second thermal. There was a second maximum at z = 7 – 9 km (e.g., Figure 12).

There  were very little differences in the graupel concentrations between the *multi-thermals* run and the *control* run before 45 min by comparing Figures 6a and 10a. However, two maxima appeared at 60 min and 75 min between 7 and 8 km with the multi-thermal run. The graupel concentrations  were much higher, 12 and 17 L$^{-1}$, compared to less than 2 L$^{-1}$ at about 40 min in the *control* run, which indicated more riming in the *multi-thermals* run. Similarly, there  were very little changes in the ice crystal concentration before 45 min. Associated with the two graupel maxima, there  were two ice crystal maxima: one maximu being 84.9 L$^{-1}$ at 58 min and z = 6.9 km and the other maximum being 121 L$^{-1}$ at 69 min and z = 6.75 km.

Figure 12 presents a time sequence of cloud properties for the THERMAL run between 54 – 69 min at intervals of 3 min spanning a period of a maximum of the secondary ice production in the HM zone.

A turret containing graupel reached just above 8 km at x ≈ 7 km with strong updraught up to 7.6 ms$^{-1}$ and raindrops below the turret at 54 min. The turret developed to a slightly higher level at 57 min and started to collapse, but strong updraughts of 7.8 ms$^{-1}$ below the turret still supported the graupel particles from falling into the HM zone. As the turret continued to collapse and the updraughts below the turret weakened, graupel particles fell down into the HM zone at 60 min. A local maximum of ice concentration appeared around 7 km in altitude and x ≈ 7.2 km, which was produced by the HM process. During the next 9 min, the coexistence of graupel particles, drops and raindrops produced secondary ice particles in the zone to 121 L$^{-1}$. More riming and hence secondary ice particles were produced in the second thermal due to the increase in liquid water content in the updraught.

The results are consistent with the findings of Blyth and Latham (1997) in that multi-thermals can significantly enhance the secondary ice production. A conceptual representation of the kinematics was used in the detailed microphysics model described by Blyth and Latham (1997), whilst the present study employed a cloud model with detailed cloud microphysical processes. There have been a few studies of thermals in shallow convective clouds (e.g., Heus et al., 2009; Heiblum et al., 2016). It is impossible to make a direct comparison of thermals between the deep convective cloud in this paper and those shallow clouds, but similar features were found, such as enhanced vertical velocities and cloud mass associated with the thermals. The injection time of 20 min was chosen when the updraught was about to decay (Figure 11 ). An earlier injection time (e.g., 10 min) of the second bubble only slightly increased the first main updraught and did not change the result significantly.

## 5. Discussions

Firstly, we discuss if the conditions for the rime-splintering process were met in cloud penetrations. Figure 13 shows the variations of the aircraft measured vertical velocity and the cloud drop concentration at 6180 m between 15:12:07 and 15:15:39 UTC. The vertical velocities indicate the weak updraught in the cloud was surrounded by downdraughts. The vertical velocities were approximate 2 ms$^{-1}$ at the time when columns were measured (Figure 14). The concentrations of drops in the cloud were a few tens per cubic centimetre (Fig. 13c). In Figure 14, graupel particles, small droplets and large drops were observed during the same pass as in Figure 13. Fragments of frozen drops were found, but not in a great amount, compared with the amount of columnar crystals, although the exact concentration needed to be determined. The high concentrations of ice particles were most likely produced by the rime-splintering process because all the conditions for the process were met. However, there is no causal evidence of other mechanisms of secondary ice production. Qu et al. (2020), for example, showed that several SIP mechanisms can operate within a convective cloud. Evidence for other secondary ice mechanisms could be provided by cloud particle images. For example, images of fragmented ice, especially pieces of dendrites are related to the ice-ice collision mechanism. Images of ice with irregular shapes are most likely produced by the drop shattering during freezing  (Field et al., 2017).

Another question is that whether very active (e.g, biogenic) INPs existed in the environment of the observed clouds (Lloyd et al., 2020). As described in Section 2, the aerosol particles in the lowest 1.5 km were mostly marine with some polluted dust, and the upper layer at 2 – 3 km consisted mostly of dust. The laser ablation aerosol particle time-of-flight mass spectrometer (LAAPTOF), was deployed to measure aerosol properties during the field campaign. The organic-biogenic fractions were moderately high in the measured dust particles at the Praia airport on 21 August (Marsden et al., 2018), which could affect the ice formation in the clouds. Although the ground measurement site was about 150 km away from the clouds, it is possible that aerosol particles in the aircraft operation region had similar chemical composition to those in the ground site.

It is noted that other mechanisms of secondary ice production may have operated in these clouds, such as fragmentation during evaporation (Bacon et al. 1998), crystal-crystal collision (Takahashi et al., 1995), fragile needles combined with ice–ice collision fragmentation (Knight, 2012), shattering following the freezing of supercooled raindrops (Leisner et al., 2014, Wildeman et al., 2017). The shattering mechanism may be most efficient between -10 ºC and -15 ºC, and ice fragments generated by shattering may be transferred to lower or higher altitudes due to the updraughts and downdraughts. The Knight mechanism (Knight, 2012) operates at a similar temperature range, and future studies can investigate the relative importance of this mechanism using new parameterisations accounting for the ice–ice collision processes (e.g. Phillips et al., 2017). Since the tops of the clouds in this study were no higher than -10 ºC and the conditions in the clouds were conducive to the HM process as currently understood, we have focussed on modelling the production of ice particles by that rime-splintering process. The recent development of the parametrization of secondary ice particles from frozen drops by Phillips et al. (2017) indicates that this process might have a considerable contribution to the secondary ice formation at temperatures greater than -10 °C. Future research will undoubtedly include this parametrization. It should also be noted that research continues on mechanisms that can cause an enhancement of ice particles (e.g. James et al., 2021).

Figure 5a shows that the cloud model produced several separate thermals over the approximately 4-km width of the cloud system. Only one of them (between 6 -7 km) developed and ascended to 9 km. As discussed by Heus et al. (2009), the inflow of air from the subcloud thermal is assumed to be in balance with detrainment from the cloud into the environment in a mature cloud. In a single cloud simulation, a cloud is triggered by perturbations in temperature as a thermal. Since the lower boundary conditions were prescribed and only random perturbations were added, there were no subsequent thermals to produce profound heat or momentum fluxes from the underlying surface that were strong enough to produce more new cloud droplets near the cloud base. Recently, Heiblum et al. simulated the cumulus field using a LES model and showed a series of in-cloud positively buoyant thermals spanning 5 – 15 min each, in precipitating clouds (their Figure 3). However, there was only one thermal in the non-precipitating cases. Their results indicated that their multi-thermals could be a result of cold pool interaction and subsequent lifting. The multi-thermal in convective clouds could be topographically or thermally forced in the mountainous region. In principle, NWP or cloud models will be able to describe the appearance of sequential thermals if the boundary layer conditions are realistically represented. This study and previous studies by Ludlam and Scorer (1953),

Koenig (1963), and Mason and Jonas (1974) have highlighted the importance of atmospheric models being able to simulate these entities.


The behaviour of a simulated cloud is sensitive to the initial conditions. A perfect initialisation requires following the trajectory of the cloud in time and space to get the vertical variation of thermodynamic variables before its formation. The initial conditions we could get in close proximity to reality was from the aircraft profile run after it took off to reach the clouds. The initialisation based on the combination of aircraft measurement and reanalysis was a source of uncertainty.

However, the major conclusion of this study is that the multi thermal is the only way to get enough ice.  The initial conditions only have to be roughly correct to get a cloud that goes to the correct height and has the same magnitude of updraft velocities and horizontal extent. Initial conditions not only affect the thermodynamic environment of a cloud but directly affect ice nucleation and rime splintering. Therefore, it is important to use the aircraft measurement of temperature and moisture profiles close to the clouds to be simulated.

**5. Conclusions**

Numerical simulations of the 21 August 2015 ICE-D deep convective clouds in the Cape Verde region examined the secondary ice production through the riming-splintering process and the sensitivity to the onset freezing temperature or/and freezing efficiency, as well as the impact of multiple thermals.

CM1 was run for the 21 August case. The default Morrison microphysics scheme was applied for the simulations. Additional simulations were run with adjusted onset freezing temperature, freezing efficiency, and the Paukert scheme for dust alone INPs, and a two-thermal simulation. The *control* simulation produced a maximum concentration of secondary ice a few per litre in the HM temperature zone which is much lower than the observed value. One possible reason for the underestimation is that the default onset freezing temperature is -8 ºC, which means the first ice in the *control* run appeared too late at higher

levels. Relaxing the onset temperature from -8 to -3 ºC doubled the maximum concentration of the secondary ice, but is still not as high as observed. Enhancing the freezing efficiency made more primary ice particles and tripled the maximum concentration, which is still not enough. Some combinations of the relaxation and enhancement lead to higher concentrations of secondary ice close to the observations. The results suggest that more active INPs with higher onset freezing temperatures are needed to produce similar amounts of secondary ice. It is possible that INPs in this case might include k-rich feldspar

aerosol particles or/and dust particles attached with biogenic INPs of high onset freezing temperatures, such as bacteria.

The simulations with the *Paukert* schemes to consider the cases with dust alone INPs  resulted in secondary ice production even lower than the *control* simulation. Because the onset freezing temperature in the *Paukert* scheme is lower than in the Morrison scheme, the primary ice appeared late at a higher altitude, which resulted in reduced graupel in the HM zone, and

hence, decreases in the secondary ice production. Dust INPs alone are not enough to produce similar amounts of secondary ice.

In the multiple thermal run, the secondary ice production associated with the first thermal is identical to that in the *control* run. However, the secondary ice concentration associated with the second thermal is much higher. The reason for the much
more secondary ice is because the second thermal produces freshly formed drops and raindrops in the HM zone for graupel particles to collect and promotes the riming-splintering process. The cloud dynamics is also important in secondary ice production. The multiple thermal simulation still cannot produce the highest observed ice number concentration.

A thorough explanation of the secondary ice production faces several challenges. The first challenge is the measurement of
secondary ice particles, particularly the small newly-formed ice particles. Instruments have typically been unable to distinguish small ice particles from cloud drops. However, there are new instruments, such as the HALOHolo (Lloyd et al., 2020) that offer some hope for making progress. The second is the challenge in measuring the full spectra of all INPs and CCN (Kanji et al., 2017). The third is the identification of mechanisms of secondary ice production (Field et al., 2017). The fourth challenge is the dynamics and the interaction with cloud microphysics (Field et al., 2017). The development of new
instruments and sampling techniques will help improve our understanding of secondary ice production.

**Code availability:** The CM1 model is available at https://www2.mmm.ucar.edu/people/bryan/cm1.

**Author contribution:** ZC designed the experiments and performed the simulations. GL analysed the ice particle concentrations. ZC and AB prepared the manuscript with contributions from all co-authors.
**Competing interests**: The authors declare that they have no conflict of interest.

**Acknowledgements:** This project was funded by UK's Natural Environment Research Council (NE/M00340X/1). NCEP Reanalysis data were provided by the NOAA/OAR/ESRL PSD, Boulder, Colorado, USA, from their Web site at https://www.esrl.noaa.gov/psd/. We are grateful to Dr. George Bryan for his development and maintenance of CM1. The BAe-146 aircrew and instrument operators are greatly appreciated. ZC is partly funded by The Centre for Environmental
Modelling And Computation (CEMAC), University of Leeds.

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

| Experiment | Description |
|---|---|
| *control* | The control run using the Morrison scheme |
| *double-HM* | Same as *control*, except the rate of the HM process is doubled |
| *early onset1* | Same as *control*, except the onset freezing temperature being -3 ºC, rather than -8 ºC |
| early onset1 & noHM | Same as *control*, except the onset freezing temperature being -3 ºC, rather than -8 ºC, but switch off the HM process |
| *Cooper10x* | The ice nuclei number concentration from the Cooper scheme is multiplied by 10 |
| early onset1 & *Cooper10x* | Combination of early onset1 and *Cooper10x* |
| early onset1 & *100xINP* | The onset freezing temperature being -3 ºC and the ice nuclei number concentration is multiplied by 100 |
| *early onset2* & *100xINP* | The onset freezing temperature being -2 ºC and the ice nuclei number concentration is multiplied by 100 |
| *Demott* | Use DeMott et al. (2010) best fit for all data, $0.117 \exp(-0.125*(TK - 273.2))$, with the onset freezing tempearature being -8 ºC |
| *early Demott* | Same as *Demott*, but the staring freezing Temperature is -3 ºC |
| *Demott 10xINP* | Same as *early Demott*, but the ice nuclei number concentration is multiplied by 2 |
| *Paukert* | Paukert scheme |
| *Paukert-dust* | Same as *Paukert*, but the ice nuclei number concentration is enhanced in the dust layer |
| *multi-thermals* | A second bubble of 2 ºC is imposed at 20 min from simulation |

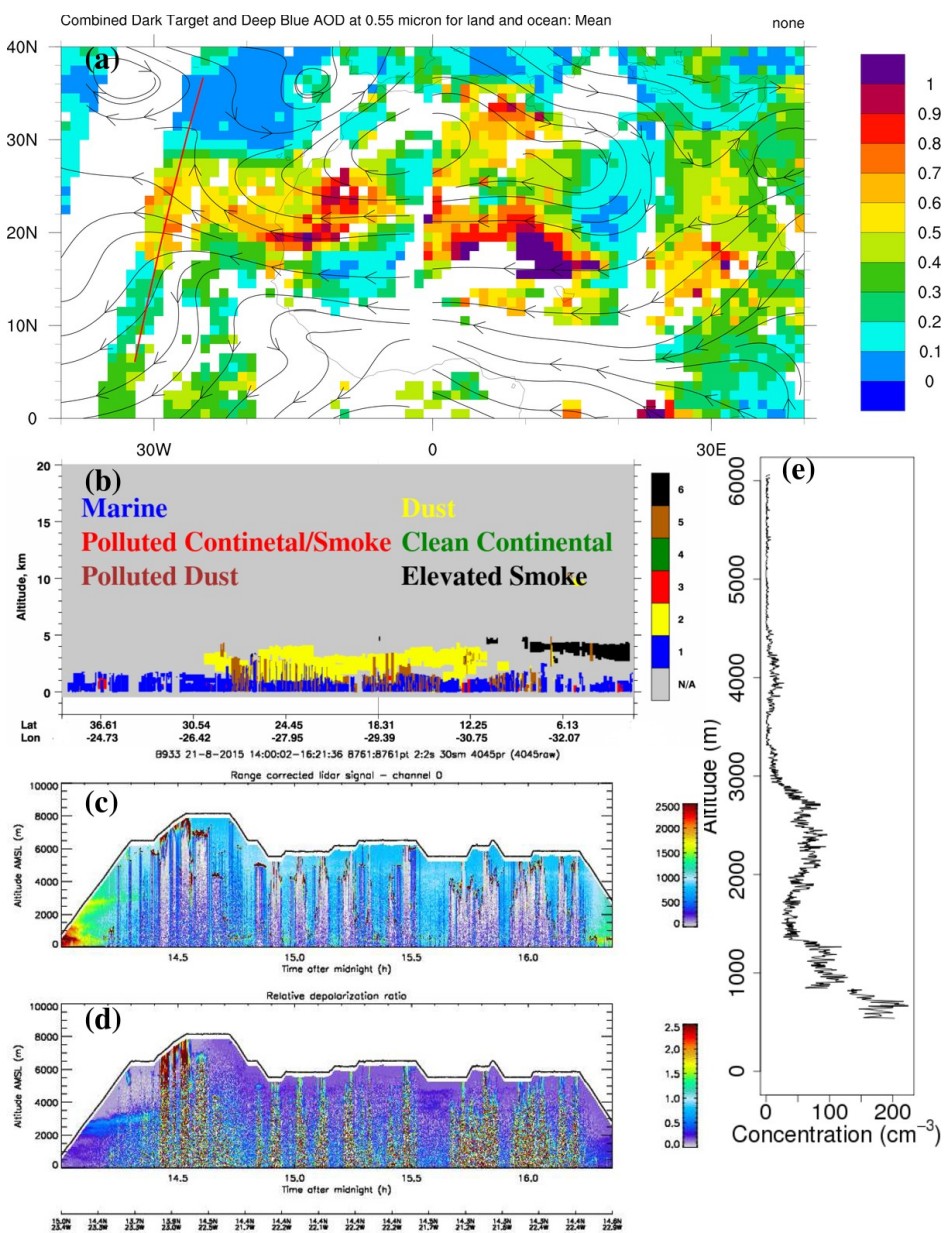


**Figure 1. (a) MODIS combined Dark Target and Deep Blue mean AOD at 0.55 µm for land and ocean on 21 August 2015, with the red line representing the path of the CALIPSO, (b) CALIPSO aerosol subtypes on 21 August 2015, (c) range correlated lidar signal and (d) relative depolarization ratio measured with the UK Met Office's Lidar on board the BAe 146 aircraft, (e) vertical variation of aerosol concentration measured with the PCASP on board the BAe 146 aircraft.**




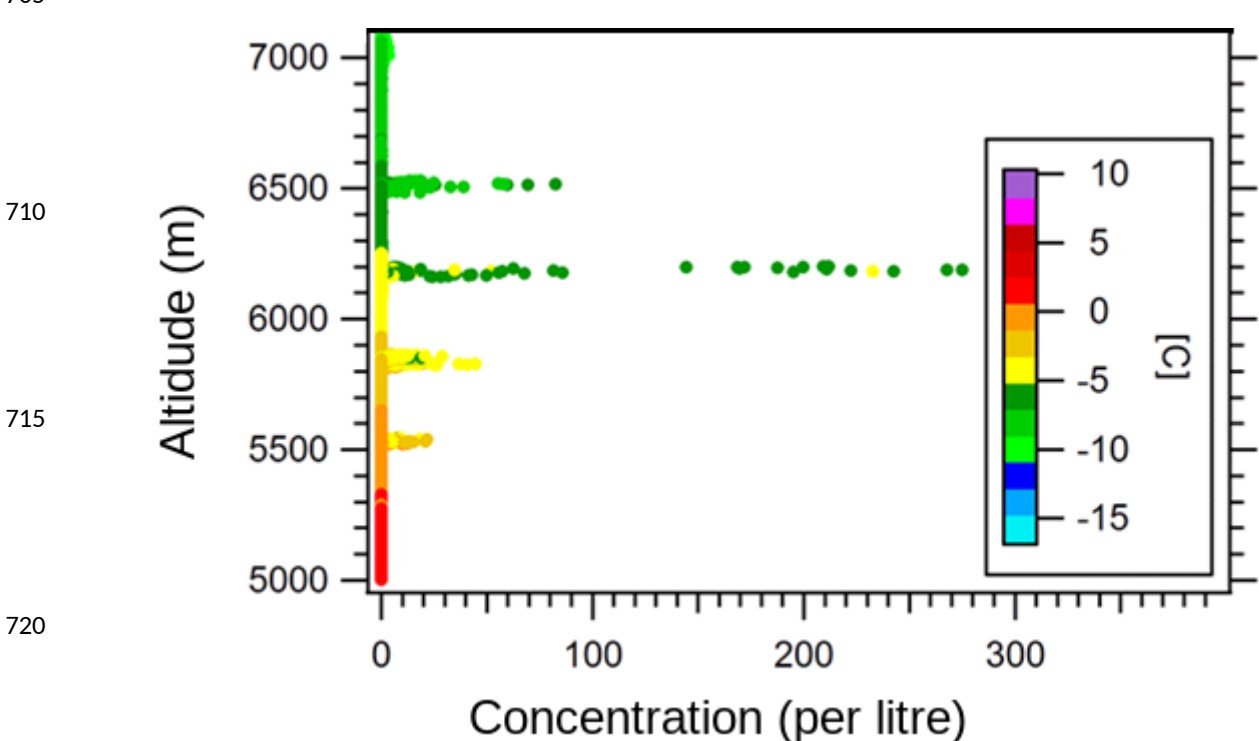




**Figure 2. Ice concentration as a function of altitude as measured with the 2DS probe on board the Bae 146 aircraft. The colour bar represents temperature.**


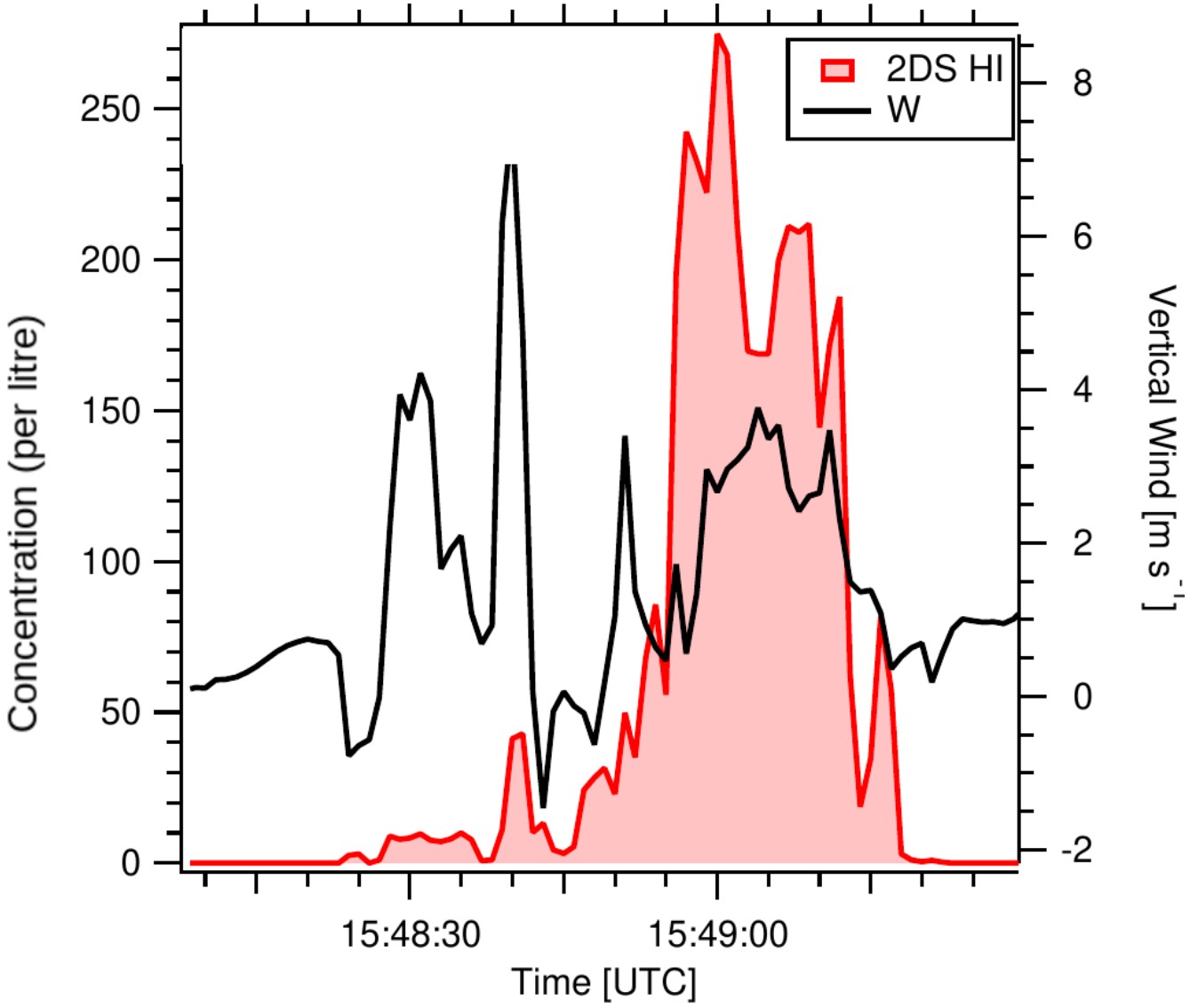

**Figure 3. Time series of the concentration of ice particles (L$^{-1}$) in the size range of 50 -1280 μm measured with 2D-S Stereo *Probe* and vertical velocity (ms$^{-1}$) between 15:48:05 UTC and 15:49:30 UTC on 21 August 2015.**

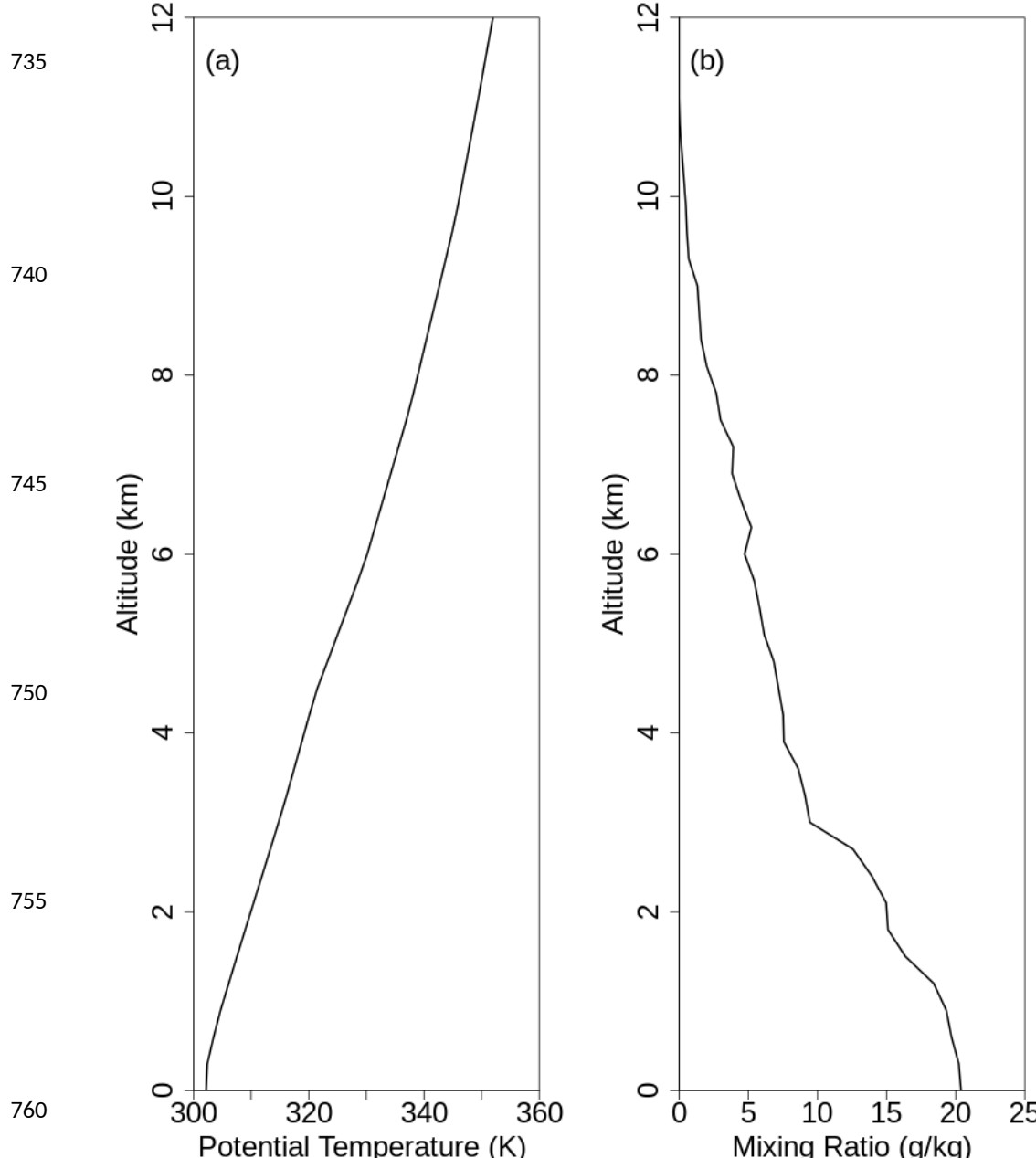

Figure 4. Initial profiles of (a) potential temperature and (b) mixing ratio for the model simulations. Altitude is the height above mean sea level.

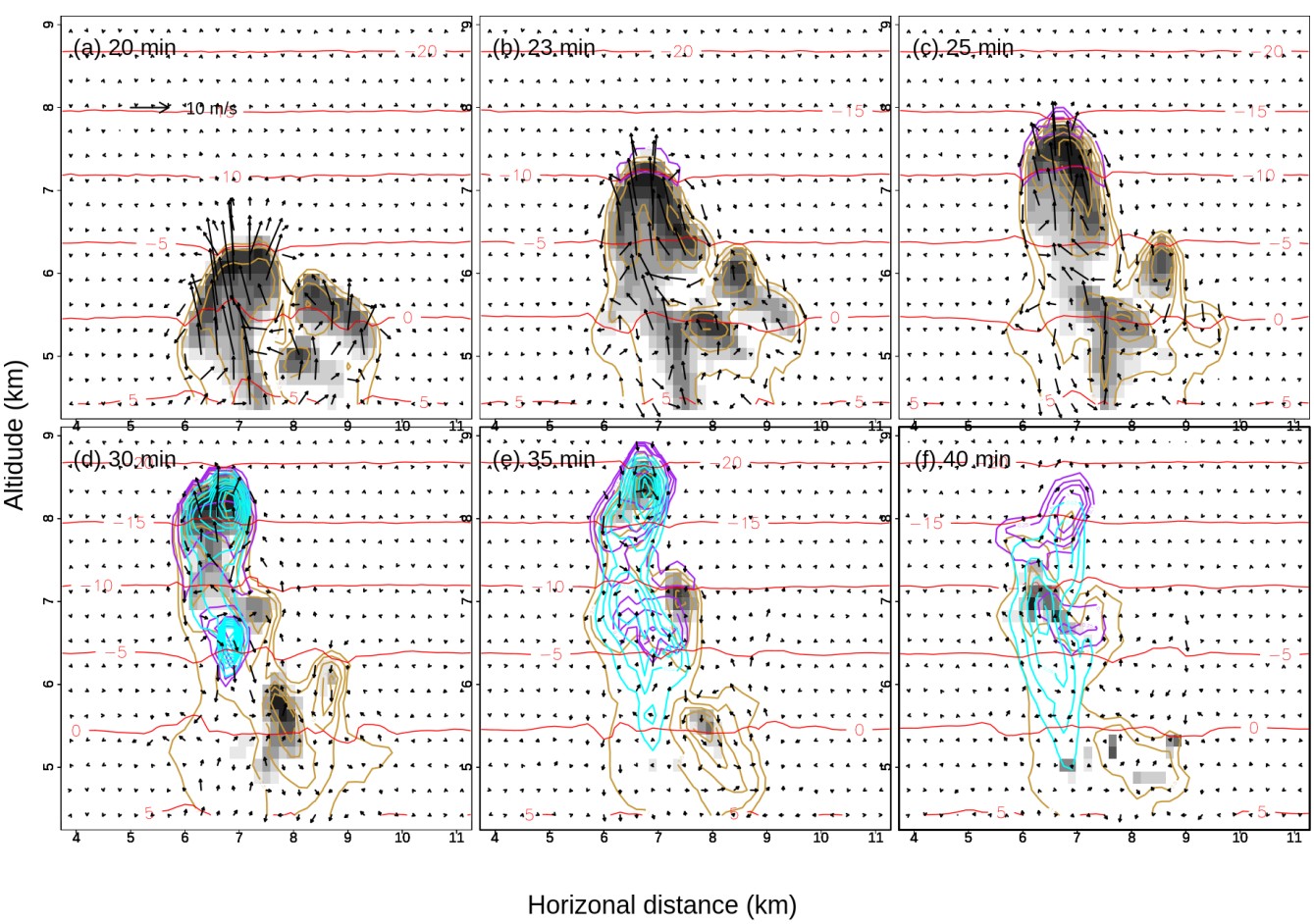

**Figure 5.** Time sequence for the control run control of spatial distribution of wind vectors, concentration of raindrops, ice crystals, and graupel particles at (a) 20 min, (b) 23 min, (c) 25 min, (d) 30min, (e) 35min and (f) 40 min. The orange, puple and cyan lines are the concentration of raindrops (contours at 1, then 2.5 to 10 in intervals of 2.5, then in intervals of 5 to 70 L$^{-1}$), ice crystals (contours at 0.25, 0.5, 1, 2.5, 5, 7.5, 10, 12.5, 15, 17.5, 20, 22.5L$^{-1}$), and graupel particles (contour at 0.02L$^{-1}$), respectively. The shade represents cloud drop mixing ratio in each panel. The x-axis and y-axis are distance (km) and altitude (km), respectively. Also shown in each panel are the maximum concentrations and scale of the wind vectors. Red lines are temperature in ºC.

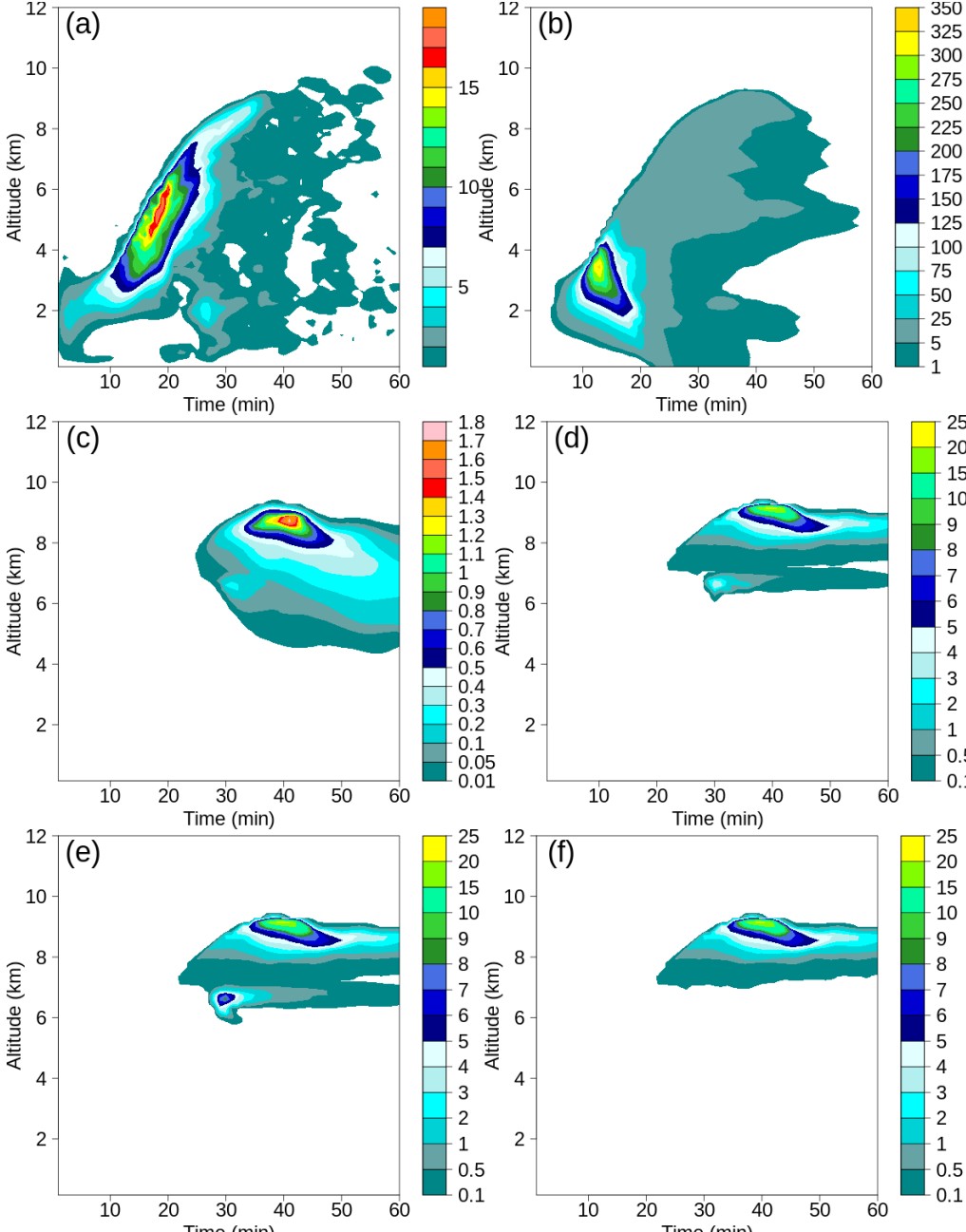

**Figure 6. Time-height variations of maximum values in (a) vertical velocity, (b) raindrop concentration, (c) graupel particle concentration,(d) ice crystal concentration in the control run (control), as well as (e) ice crystal concentration in the early onset1 run, and (f) ice crystal concentration in the early onset1 & noHM run.**

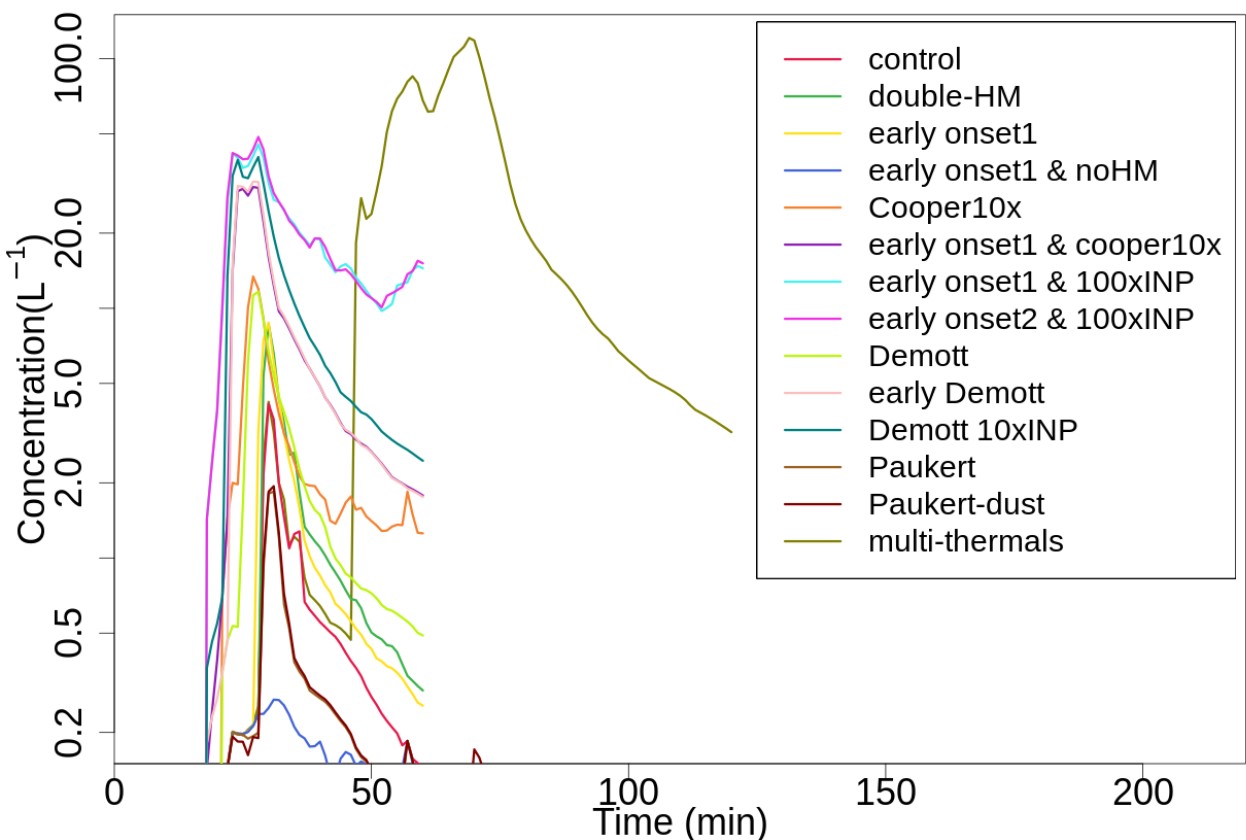

**Figure 7. Temporal variation of maximum concentrations of ice crystals in the Hallett-Mossop temperature zone**

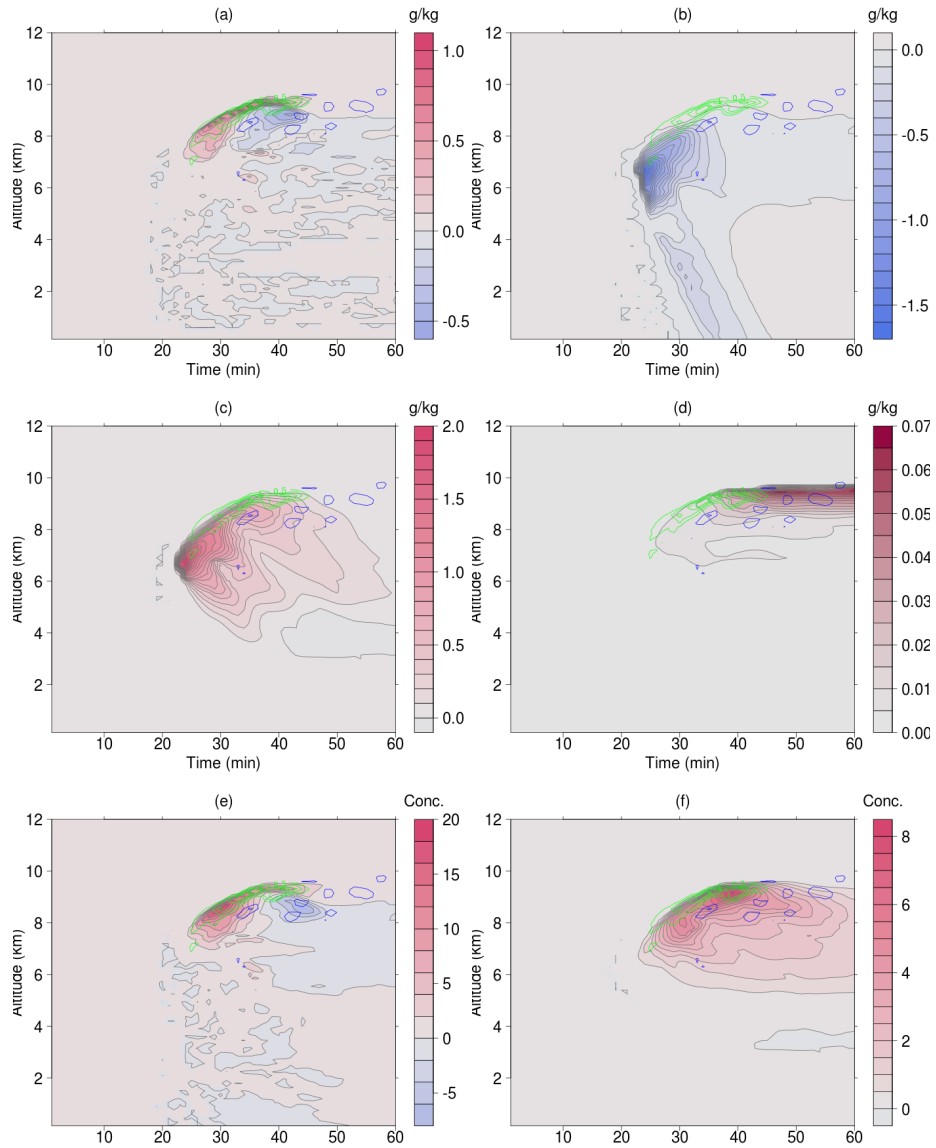

**F**

**Figure 8. Difference between the sensitivity runearly onset1 & Cooper10x, and the control run, control: (a) cloud water mixing ratio (gkg⁻¹), (b) rain water mixing ratio (gkg⁻¹), (c) graupel mixing ratio (gkg⁻¹), (d) ice mixing ratio (gkg⁻¹), (e) raindrop concentration (per litre), and (f) graupel concentration (per litre), respectively. Imposed on these figures are the differences in vertical velocity, with green contour lines indicating positive and blue contour lines negative differences, and contour intervals of 0.5 ms⁻¹.**

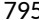

**Figure 9. Difference in the ice crystal number concentrations (per litre) between sensitivity run and the control run: (a) early onset1 , (b)** *Cooper10x***, (c) early onset1 &** *Cooper10x* **, (d) early onset1 &** *100xINP***, (e)** *early onset2 &* *100xINP***, (f)** *Demott***, (g)** *early Demott* **and (h)** *Demott 10xINP* **, respectively. The x-axis and y-axis are time (min) and altitude (km), respectively.**

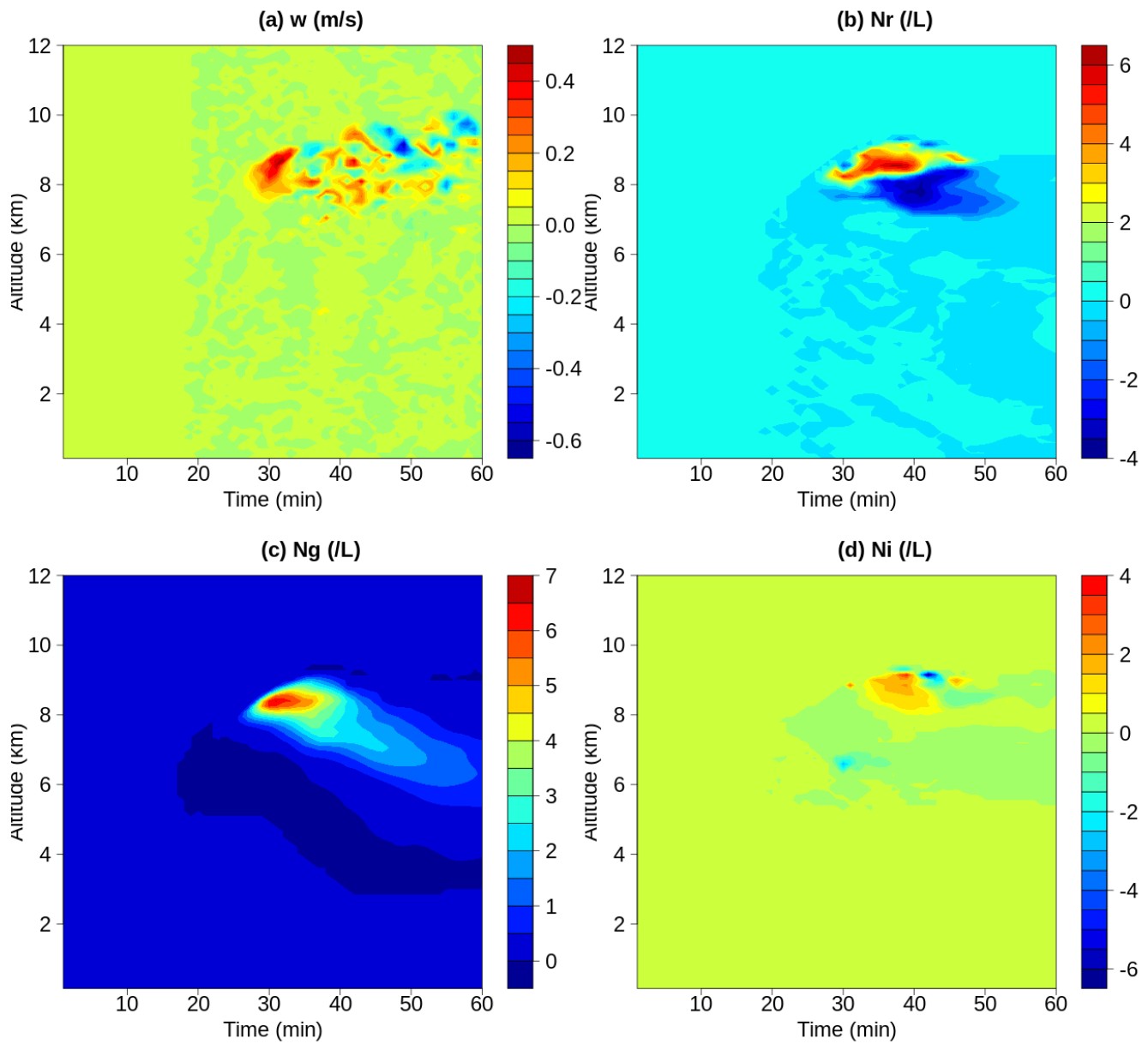

**Figure 10. Difference between the *Paukert* run and the control run: (a) vertical velocity (ms⁻¹), (b) raindrop number concentration**
**(per litre), (c) the graupel concentrations (per litre), and (d) the ice crystal concentrations (per litre), respectively.**

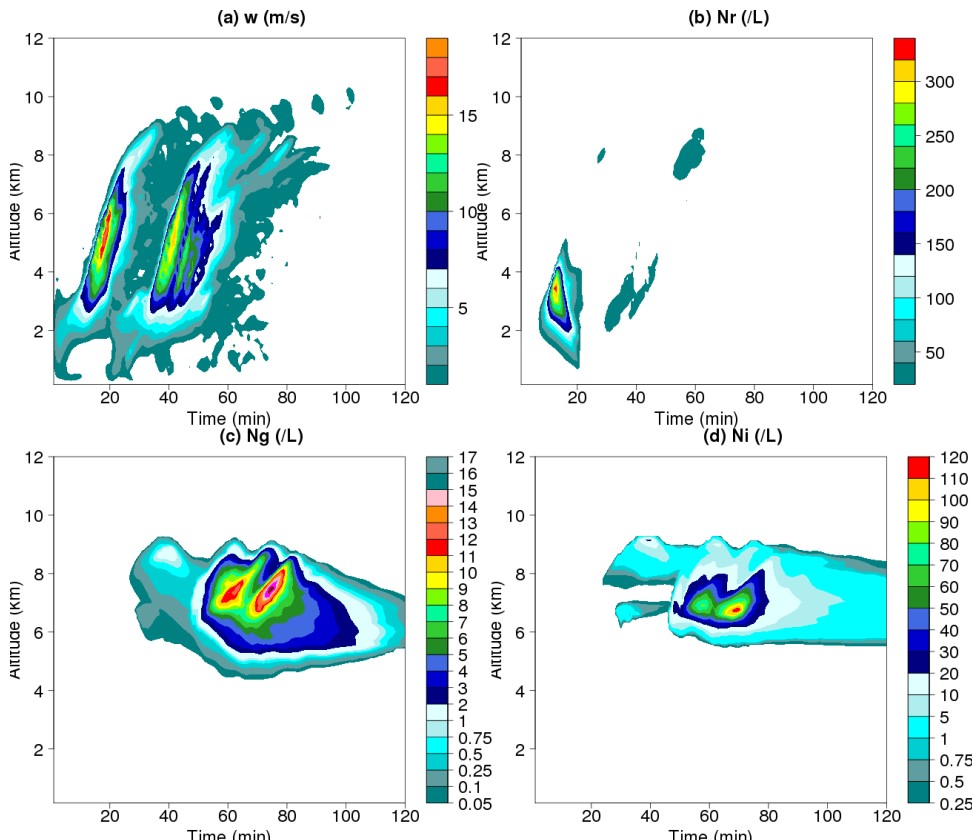

**Figure 11. Time-height variations of maximum values in vertical velocity (a), raindrop concentration (b), graupel particle concentration (c), and ice crystal concentration (d) in the two thermal run (*multi-thermals*).**

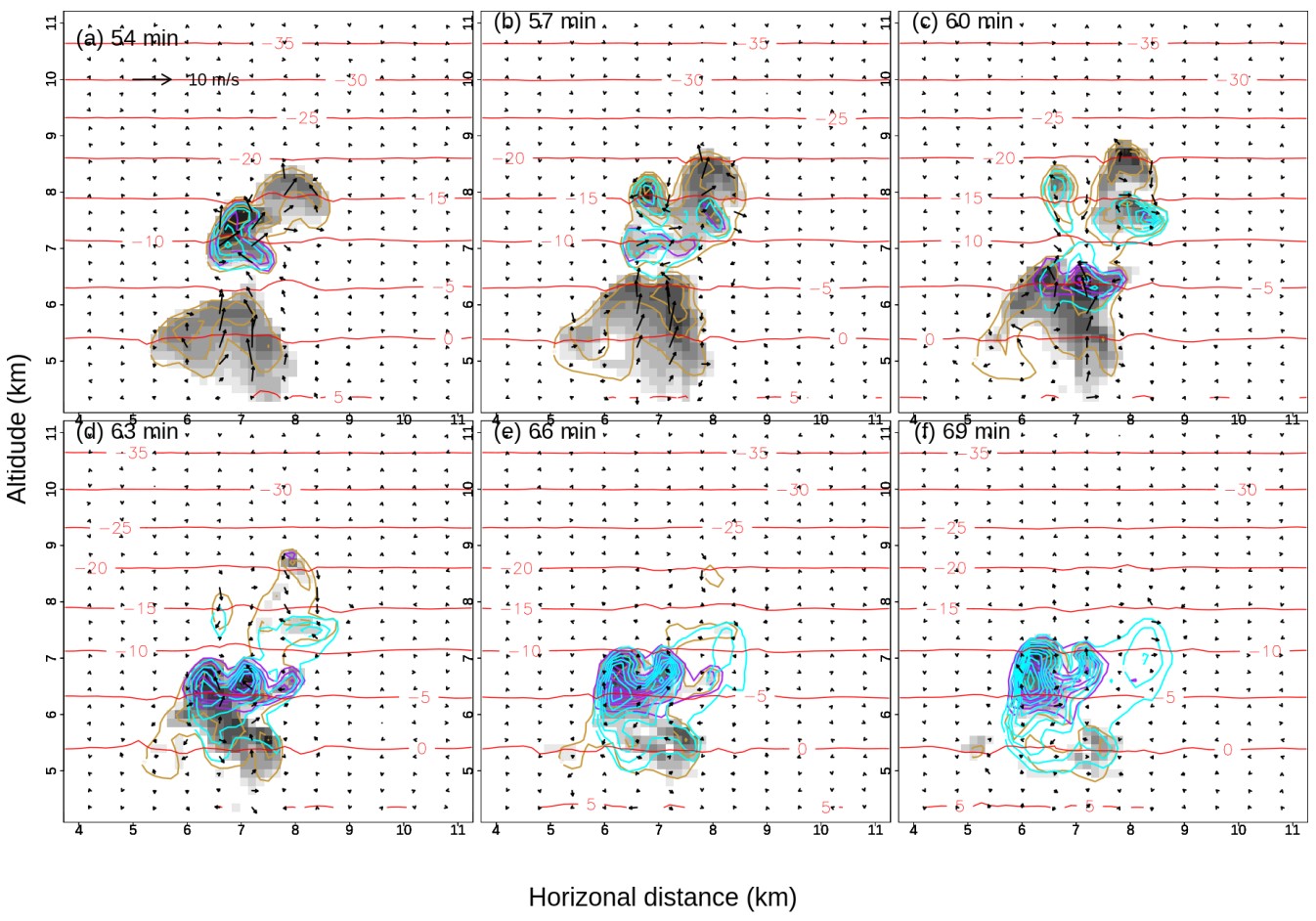

**Figure 12. Time sequence for the** *multi-thermals* **run of spatial distribution of wind vectors, concentration of raindrops, ice crystals, and graupel particles at (a) 54 min, (b) 57 min, (c) 60 min, (d) 63 min, (e) 66 min and (f) 69 min. The orange, pupple and cyan lines are the concentration of raindrops (intervals at $2L^{-1}$), ice crystals (intervals at $10L^{-1}$), and graupel particles (intervals at $2L^{-1}$), respectively. Cloud drop mass mixing ratio is shown in shade in each panel. The cyan lines are isotherms in ◦C. The maximum concentrations are shown in each panel in $L^{-1}$. The scale for the wind vectors is also shown in (a).**

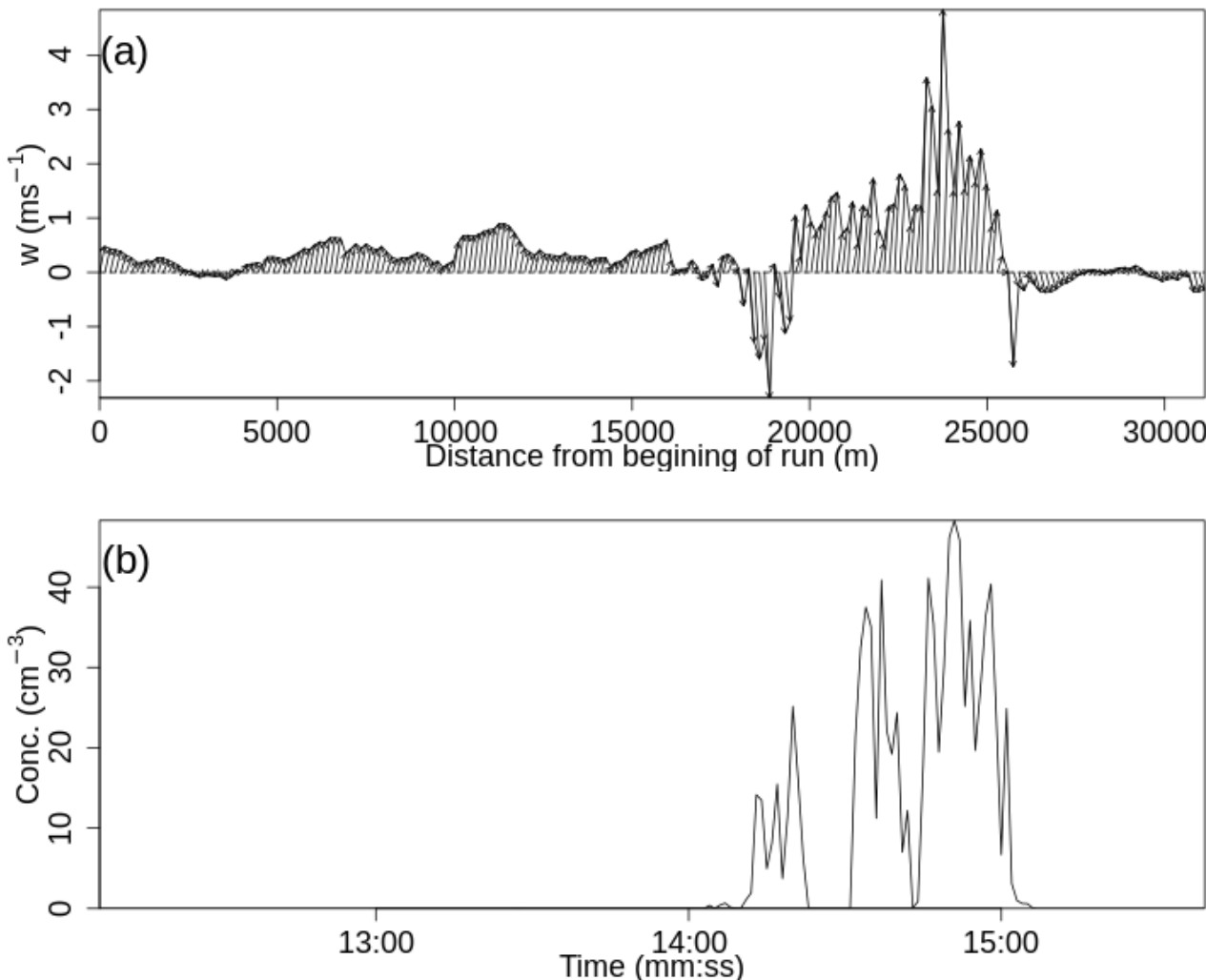

**Figure 13. Variations of the vector winds (a) and cloud drop concentration measured with the CDP (b).**

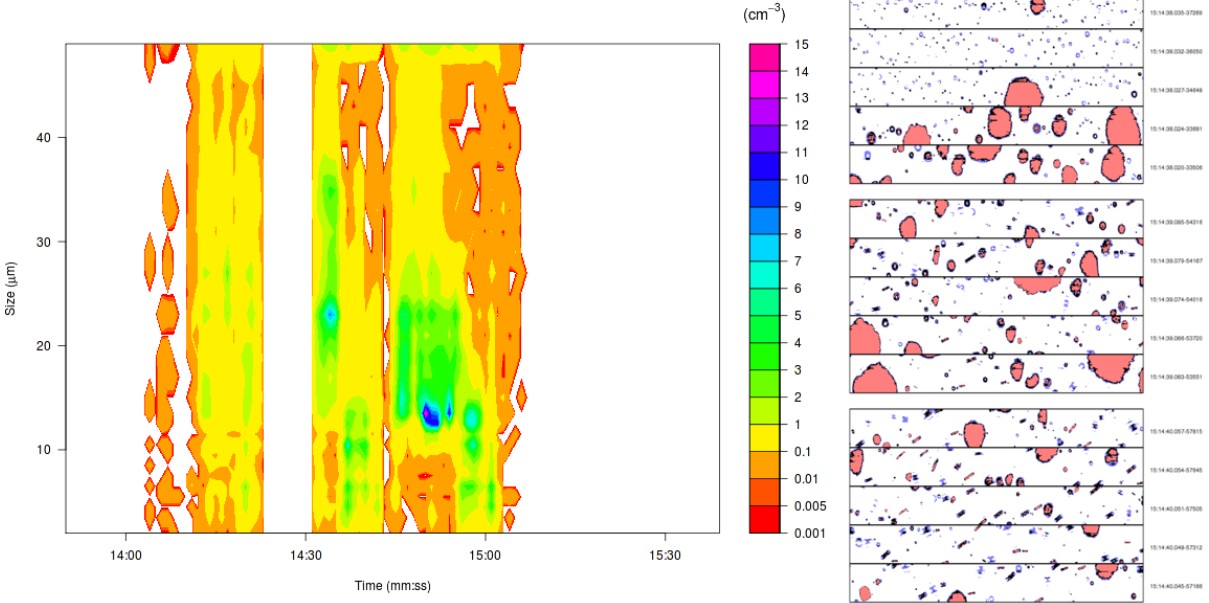

**Figure 14. (left): Size distribution of drops measured with the Cloud Drop Probe, and (right) examples of images measured with the Cloud Imaging Probe (CIP) during Run 6. The CIP image width is 960 µm.**