# Peer review of "Multi-thermals and high concentrations of secondary ice: A modelling study of convective clouds during the ICE-D campaign"

_Atmospheric Chemistry and Physics, 2021_

## Referee Comment (RC1)

REVIEW FOR:

**Multi-thermals and high concentrations of secondary ice: A modelling study of convective clouds during the ICE-D campaign**
*by Cui et al.*

This paper examines the mechanisms responsible for enhanced ice crystal number concentrations (ICNCs) in convective clouds observed during ICE-D campaign. Sensitivity tests are performed through variations in the primary ice nucleation scheme and the dynamic conditions to investigate their impact on secondary ice production through rime-splintering. Changes in the onset freezing temperature or the freezing efficiency of the nucleation scheme result in weak ICNC enhancement, not sufficient to explain observations. The highest concentrations are achieved by inducing a second instability in the domain to generate an additional thermal. The authors conclude that multi-thermals play a critical role in generating large ICNCs in tropical clouds.

**Major comments:**
The subject of this study is very interesting as the mechanisms responsible for the large ICNCs in convective clouds remain poorly understood, and especially those that concern secondary ice production. However, due to lack of detailed description of the sensitivity simulations it is not very easy to derive any robust conclusions. For example, the authors mention that Cooper parameterization is used in the Morrison scheme to treat primary ice production. However, as far as I know, this scheme utilizes different expressions for different freezing mechanisms (condensation, immersion, contact) and Cooper represents only one of them. Is this indeed the case with the particular model? If yes, then it doesn't make much sense to modify the freezing efficiency or onset temperature for only a single freezing parameterization (e.g. while the onset temperature for immersion freezing remains constant at -4$^{o}$C as indicated in Morrison et al. 2005). Also it wouldn't make much sense to replace the Cooper description with the immersion freezing parameterization by Paukert and Hoose (2014), if the immersion mechanism is represented by Bigg (1953) formula in the scheme. This means that the immersion freezing process would be described twice in the respective simulations.

Finally, since the main conclusion of the study is based on the THERMAL simulation, it is important to understand the realism of this experimental set-up. While injecting instabilities at the beginning of Large-Eddy or Cloud-Resolving Simulations is necessary, I would expect that the model would eventually be able to develop turbulent motions in a relatively realistic way. So basically I don't understand why a high-resolution model cannot produce multi-thermals in a 15 km x 15 km domain. I would appreciate a discussion on the ability of cloud models and weather prediction models to represent such turbulent structures. This would highlight the importance of this study's findings for atmospheric modeling.

**Minor comments:**

**Line 26:** please specify… enhancement of **the freezing efficiency**… Relaxation of the **onset temperature**

**Lines 29-30:** *in a similar way to other convective clouds observed elsewhere in the world'* This is a general statement that is not suitable for the abstract section, which aims to summarize main results. You have not provided references in the text for studies that have shown that multi thermals are important for cloud ice generation in 'other convective clouds elsewhere in the world'

**Line 52:** are you sure about the $-14^o$C threshold? I think Wilson et al (2015) showed that marine organics in sea-spray aerosols can nucleate at temperatures up to $-10^o$C in the immersion mode

**Line 63:** *The INP concentrations are highly variable at a temperature.* This is a very general statement. You mean variable at a given temperature depending on aerosol, dynamic and humidity conditions?

**Line 73:** Lawson et al (2015) developed a parameterization for secondary ice. Lawson et al. (2017) developed an expression that predicts the level in the updraft core, where liquid water gets depleted.

**Line 78-79:** you might want to rephrase this. There are actually models that are fully coupled with a chemistry component. Maybe say that this is not the case for most models?

**Lines 91-95:** *'It occurred before…recommendation'.* This sentence is redundant. It is common to analyze older campaign data to address different scientific questions.

**Line 95:** *'previous field campaigns relevant to ice nucleation'.* Maybe say 'campaigns investigating or focusing on ice nucleation'? Anyway, this needs to be rephrased

**Line 105:** *'Secondly, the first ice particles appeared at temperatures greater than -8 °C.'* Why is this mentioned here as a surprising finding? As you state in the introduction, there are several dust types that nucleate at temperatures higher than $-10^o$C. Is this because such aerosol types are not expected in the studied region? Please explain.

**Line 125:** *'to measure… retrieval of aerosol optical properties'* Please rephrase, it does not make any sense. You mean 'to retrieve aerosol optical properties'?

**Line 145:** *'Only the outside-cloud temperatures were used to avoid the bias caused by wetting'…* were used for what?

**Lines 152-153:** *'Firstly, the ice particles appeared at temperatures greater than -10 $^o$C'.* Please expand the discussion why this is listed as a main point of the study. Ice presence at such temperatures has been observed many times in dust-dominated environments. Even if

we exclude nucleation, seeding from higher levels is also a possibility

**Line 155:** *'Cloud … designed for idealised simulation of convective clouds using relatively less memory'* Could you provide information on the adapted simplifications to understand the level of idealization in the simulations? This would help assess the realism of the results, especially in terms of the simulated dynamics

**Line 158:** maybe say 'which allows for quick simulations'

**Section 3.2:** Please provide a detailed description of the individual parameterizations and sensitivity set-ups. The DeMott expression and simulation description is only stated in the Table. Also please provide details about the Paukert and Hoose (2014) parameterization. For example, it is not mentioned until the Results section that a colder onset temperature is used in this description. Also later you mention that you modified the INP number to drop number ratio in this scheme, without giving any information on the default values or the implemented modifications. Describing the set-up of each sensitivity test in detail is critical for the interpretation and reproduction of the results.

**Line 174:** *'noted that the factor was 4'.* Please explain what is meant by this phrase. Do you mean that this is the factor that provides the best agreement with observations?

**Line 180:** *'DeMott et al. (2010) based on all available data'* . This statement is wrong. This parameterization takes into account several datasets from different regions, but definitely not all available INP data.

**Line 186:** Explain PAUKERTD set-up. How ice concentration is enhanced in the dust layer? (actually all tests should be described in detail).

**Line 194-197:** Please add a figure that shows the profiles used to initiate the simulations. Documenting the initial thermodynamic conditions is important for comparison with other modeling studies or for other modellers to simulate the same case

**Line 198:** *'where another bubble was added 20 min from the starting time'.* I am familiar with the fact that a bubble is needed to initiate instability in cloud resolving simulations, but after this I thought that the models can develop turbulence and simulate multi thermals within a 15x 15 km region. What is the physical explanation for the second bubble? What are the model limitations in representing boundary-layer dynamics? Does grid spacing has any impact on this? Also is choice of injection time random or does it have any physical basis (e.g. updraft lifetime). And do results show sensitivity to this choice?

**Line 216:** *'The cloud top continued to descend was about 8.5 km (T ~ - 18 $^{o}$C) by 40 min'* Please rephrase, it does not make any sense

**Line 217:** you refer to the ice crystal concentrations? (graupel is also ice)

**Line 221**: please fill the empty brackets

**Line 251:** What do you mean 'without the dust layer being considered'? Is the parameterization applied only up to a certain level?

**Line 261:** replace 'next' and 'later' with 'in Section…'

**Line 263:** *' due to different microphysical schemes'* , you use the same microphysical scheme (Morrison). Please rephrase

**Line 295:** maybe specify that you refer to the ice crystal concentration

**Line 306:** '*enhancing the efficiency'* of what?

**Line 314:** *'increased approximately above 8 km and decrease' , ' There was less riming',* sometimes past tense is used to describe results, and some other times present tense. Please be consistent throughout the whole manuscript

**Line 319-320:** further clarifications on the implemented modifications are needed

**Line 324*:** 'the different microphysical schemes affect* ', you use Morrison scheme in all runs.

**Line 326**: *' It would be incorrect not …'.* Remove this sentence or rephrase (e.g. 'it is important to consider… since both play a critical role in ice production', sth like this)

**Line 328:** maximum values of what parameter

**Line 335: …**lasted for a longer time, approximately 30 min  (rephrase)

**Line 340:** *'but there left some'* , please rephrase

**Lines 259-360*:** ' in the zone to 121 L^{-1}',* please rephrase

**Lines 363-371:** I suggest to add a third panel with temperature timeseries in Figure 12 and include it in the discussion here.

**Line 384:** '*The shattering mechanism may be most efficient between -10 C and -15 C',* yes but ice fragments generated by this mechanism at higher altitudes can be transferred at the lower cloud levels ,examined here, right?

**Line 385:** *'Although the Knight mechanism operates at the similar temperature range, there is no parameterization of the process'* Why the Phillips et al. (2017) parameterization does not account for this mechanism? They have developed a scheme that takes into account different ice types and habits. The one referred as 'planar' ice category in their study includes needles.

**Line 368-369:** *'Fragments of frozen drops were found, but not in a great amount (figures not shown)'.* This is important evidence and should be mentioned above, where the possible contribution from other SIP mechanisms is discussed. You cannot assess the contribution of

the drop-shattering process only from images, so your conclusion that this mechanism is insignificant is not necessarily accurate.

**Lines 370-371:** *'However, there is no causal evidence and we cannot rule out other mechanisms of secondary ice production'.* Maybe refer to Qu et al. (2020) who showed that several SIP mechanisms can operate within a convective cloud.

**Line 394:** '*The Morrison microphysics scheme was applied for the control run* ' Morrison scheme was applied to all runs. Maybe just clarify 'the default Morrison scheme'

**Technical corrections:**
**Line 117:** involved **an** (not in)
**Line 124:** to measure (not measures)
**Line 125:** remove 'so that'
**Line 128:** to measure instead of to measurement
**Lines 134-141:** it is Figure 1 you are referring to, not Figure 2
**Line 139:** I don't understand to which panel of Figure 1 you are referring to
**Line 287:** correct 'increasei7'
**Line 312:** PAUKERT **and** CTL
**Line 321:** to produce secondary ice  concentrations similar…
**Line 325:** observation**s**
**Line 344:** concentration**s**
**Line 406:** **'**resulted in even lower' instead of 'had'

**References:**
Qu, Y., Khain, A., Phillips, V., Ilotoviz, E., Shpund, J., Patade, S., and Chen, B. : The role of ice splintering on microphysics of deep convective clouds forming under different aerosol conditions: Simulations using the model with spectral bin microphysics. J. Geophys. Res. Atmos.*,* 125, e2019JD031312. https://doi.org/10.1029/2019JD031312, 2020

Lawson, R. P., Woods, S., & Morrison, H. (2015).   The Microphysics of Ice and Precipitation Development in Tropical Cumulus Clouds, *Journal of the Atmospheric Sciences*, *72*(6), 2429-2445.

---

## Author Comment (AC1)

**Response**

We thank the reviewer for reviewing our manuscript and providing such thorough, helpful and constructive suggestions. We have considered the comments carefully and have made appropriate changes to the manuscript. Please find below a detailed point-by-point response to all comments.

**Major comments:**

**1.** The subject of this study is very interesting as the mechanisms responsible for the large ICNCs in convective clouds remain poorly understood, and especially those that concern secondary ice production. However, due to lack of detailed description of the sensitivity simulations it is not very easy to derive any robust conclusions. For example, the authors mention that Cooper parameterization is used in the Morrison scheme to treat primary ice production. However, as far as I know, this scheme utilizes different expressions for different freezing mechanisms (condensation, immersion, contact) and Cooper represents only one of them. Is this indeed the case with the particular model? If yes, then it doesn't make much sense to modify the freezing efficiency or onset temperature for only a single freezing parameterization (e.g. while the onset temperature for immersion freezing remains constant at -4 C as indicated in Morrison et al. 2005). Also it wouldn't make much sense to replace the Cooper description with the immersion freezing parameterization by Paukert and Hoose (2014), if the immersion mechanism is represented by Bigg (1953) formula in the scheme. This means that the immersion freezing process would be described twice in the respective simulations.

> **Reply:** The description of the experimental design "The aims of the sensitivity simulations are summarised as follows. RLX examined the effect of active INPs at higher temperatures on secondary ice production. TEN explored the effect of more INPs. RLXTEN combined effects of the above two, while RLX3X100 and RLX2X100 probed the effect of even higher loadings of INPs. The DeMott scheme was examined in runs DMTA, DMTRLX, and DMTRLX2 and the effect of dust in PAUKERT and PAUKERTD. Finally, the effect of multi-thermals on secondary ice production was examined in THERMALS."

has been changed to

> "The Morrison scheme has several ice freezing modes, including the immersion freezing, the deposition freezing as a function of supersaturation with respect to ice, contact freezing, homogeneous freezing, and the secondary ice production by the HM process. For relaxation and enhancement sensitivity simulations, we only modified the immersion freezing mode. The aims of the sensitivity simulations are summarised as follows. early onset1 examined the effect of active INPs at higher temperatures on secondary ice production when the onset temperature was increased to -3 °C. Cooper10x explored the effect of more INPs (i.e., the freezing efficiency was multiplied by 10). early ohnset1 & cooper10x combined the effects of the above two, while early onset1 & 100xINP and early onset2 & 100xINP probed the effect of even higher loadings of INPs. The DeMott scheme (2010) was examined in runs Demott, early Demott, and Demott 10xINP. To investigate the effect of the dust as INP, the Bigg (1953) scheme was replaced by the Paukert and Hoose scheme (2014) since the Bigg scheme is for general INP types, but the Paukert and Hoose scheme considers different INP types. The Paukert run used the mineral dust parameters outlined in the Paukert and Hoose scheme. The Paukert-dust run was same as the Paukert run except that the INP numbers were increase by a factor 3.3 in the layer between 2 – 3 km where the dust layer was observed (Figure 1e). Finally, the effect of multi-thermals on secondary ice production was examined in multi-thermals when a second bubble of 2 °C was added after 20 min into the simulation."

**2.** Finally, since the main conclusion of the study is based on the THERMAL simulation, it is important to understand the realism of this experimental set-up. While injecting instabilities at the beginning of Large-Eddy or Cloud-Resolving Simulations is necessary, I would expect that the model would eventually be able to develop turbulent motions in a relatively realistic way. So basically I don't understand why a high-resolution model cannot produce multi-thermals in a 15 km x 15 km domain. I would appreciate a discussion on the ability of cloud models and weather prediction models to represent such turbulent structures. This would highlight the importance of this study's findings for atmospheric modeling.

> **Reply:** We added the following in the text in response to the reviewer's comment.

> "Figure 5a shows that the cloud model produced several separate thermals over the approximately 4-km width of the cloud system. Only one of them (between 6 -7 km) developed and ascended to 9 km. As discussed by Heus et al. (2009), the inflow of air from the subcloud thermal is assumed to be in balance with detrainment from the cloud into the environment in a mature cloud. In a single cloud simulation, a cloud is triggered by perturbations in temperature as a thermal. Since the lower boundary conditions were prescribed and only random perturbations were added, there were no subsequent thermals to produce profound heat or momentum fluxes from the underlying surface that were strong enough to produce more new cloud droplets near the cloud base. Recently, Heiblum et al. simulated the cumulus field using a LES model and showed a series of in-cloud positively buoyant thermals spanning 5 – 15 min each, in precipitating clouds (their Figure 3). However, there was only one thermal in the non-precipitating cases. Their results indicated that their multi-thermals could be a result of cold pool interaction and subsequent lifting. A series of thermals in convective clouds could be topographically or thermally forced in mountainous region. In principle, NWP or cloud models will be able to describe the appearance of sequential thermals if the boundary layer conditions are realistically represented. This study (and previous studies by Ludlam, Koenig, Mason and Jonas, etc) has highlighted the importance of atmospheric models being able to simulate these entities."

**Minor comments:**

Line 26: please specify... enhancement of the freezing efficiency... Relaxation of the onset temperature
> **Reply:** It has been changed.

Lines 29-30: in a similar way to other convective clouds observed elsewhere in the world' This is a general statement that is not suitable for the abstract section, which aims to summarize main results. You have not provided references in the text for studies that have shown that multi thermals are important for cloud ice generation in 'other convective clouds elsewhere in the world'
> **Reply:** The reviewer is correct. We have deleted that phrase from the sentence.

Line 52: are you sure about the -14 C threshold? I think Wilson et al (2015) showed that marine organics in sea-spray aerosols can nucleate at temperatures up to -10 C in the immersion mode.
> **Reply:** It has been changed to -10 °C.

Line 63: The INP concentrations are highly variable at a temperature. This is a very general statement. You mean variable at a given temperature depending on aerosol, dynamic and humidity conditions?

> **Reply:** It has been changed to "variable at a particular temperature that depends on the dynamics and the aerosol properties and humidity conditions."

Line 73: Lawson et al (2015) developed a parameterization for secondary ice. Lawson et al. (2017) developed an expression that predicts the level in the updraft core, where liquid water gets depleted.

> **Reply:** We added "Lawson et al (2015) developed a parameterization for the drop-freezing secondary ice production process and subsequent riming. Lawson et al. (2017) developed an expression that predicts the level in the updraft core, where liquid water becomes depleted."

Line 78-79: you might want to rephrase this. There are actually models that are fully coupled with a chemistry component. Maybe say that this is not the case for most models?

> **Reply:** It has been changed to "Many numerical models, including some cloud models, do not explicitly include the information about CCN and INP, but rather use parameterization"

Lines 91-95: 'It occurred before...recommendation'. This sentence is redundant. It is common to analyze older campaign data to address different scientific questions.

> **Reply:** The following has been deleted: "It occurred before the recommendation was made by Field et al (2017) to 'carry out integrated field programs involving in-situ sampling, remote sensing, and modelling studies'. However, the analysis of data from the project are still ongoing and hence addresses the recommendation. "

Line 95: 'previous field campaigns relevant to ice nucleation'. Maybe say 'campaigns investigating or focusing on ice nucleation'? Anyway, this needs to be rephrased.

> **Reply:** It has been changed to "campaigns that have investigated ice nucleation".

Line 105: 'Secondly, the first ice particles appeared at temperatures greater than -8 °C.' Why is this mentioned here as a surprising finding? As you state in the introduction, there are several dust types that nucleate at temperatures higher than -10 C. Is this because such aerosol types are not expected in the studied region? Please explain.

> **Reply:** Although the some dust INPs can nucleate at temperatures greater than -10 °C, we did not know whether it was the case for the clouds in this region on this day. The reasons for including those sentences are that the focus of the paper is the concentration of ice particles in convective clouds in the ICE-D region, and the observation provided us the facts of the cloud microphysics for our simulations. We added "Secondly, the first ice particles were believed to have formed at a temperature greater than about -5 °C (Lloyd et al, 2020)"

Line 125: 'to measure... retrieval of aerosol optical properties' Please rephrase, it does not make any sense. You mean 'to retrieve aerosol optical properties'?

> **Reply:** It has been changed to "and to retrieve aerosol optical properties"

Line 145: 'Only the outside-cloud temperatures were used to avoid the bias caused by wetting'... were used for what?

> **Reply:** It has been changed to "in-cloud temperatures could not be obtained because of wetting problems"

Lines 152-153: 'Firstly, the ice particles appeared at temperatures greater than -10 C'. Please expand the discussion why this is listed as a main point of the study. Ice presence at such temperatures has been observed many times in dust-dominated environments. Even if we exclude nucleation, seeding from higher levels is also a possibility

> **Reply:** "the ice particles appeared at temperatures greater than -10 C" is not a new finding. It is a fact based on the observation. We have changed the sentences: "Two points can be drawn based on the figure. Firstly, the ice particles appeared at temperatures greater than -10 C. Secondly, the ice concentrations were much higher than the predicted by the primary ice production. Therefore, secondary ice production definitely occurred in the clouds." to "The observation indicates that the ice concentrations were a few tens per litre at derived temperatures in cloud between 0 and −2 ˚C (Lloyd et al., 2020). Therefore, secondary ice production most likely occurred."

Line 155: 'Cloud ... designed for idealised simulation of convective clouds using relatively less memory' Could you provide information on the adapted simplifications to understand the level of idealization in the simulations? This would help assess the realism of the results, especially in terms of the simulated dynamics

> **Reply:** The application of CM1 has increased significantly in recent years, and papers using CM1 have been published in prestigious journals, such as Nature, ACP, JGR, JAS, MWR, JAMES, QJRMS, etc. Those sentences have been changed to "The Cloud Model 1 (CM1) was used for simulations in this study. More details on the model can be found in Bryan et al. (2003) and Bryan and Morrison (2012). The model uses conserved mass and energy conservation numerical schemes in 3-dimensions and has a rich choice of microphysics schemes. "

Line 158: maybe say 'which allows for quick simulations'
> **Reply:** It has been changed to " which allows for quick simulations"

Section 3.2: Please provide a detailed description of the individual parameterizations and sensitivity set-ups. The DeMott expression and simulation description is only stated in the Table. Also please provide details about the Paukert and Hoose (2014) parameterization. For example, it is not mentioned until the Results section that a colder onset temperature is used in this description. Also later you mention that you modified the INP number to drop number ratio in this scheme, without giving any information on the default values or the implemented modifications. Describing the set-up of each sensitivity test in detail is critical for the interpretation and reproduction of the results.
> **Reply:** Please see the reply to the Main Point 1.

Line 174: 'noted that the factor was 4'. Please explain what is meant by this phrase. Do you mean that this is the factor that provides the best agreement with observations?
> **Reply:** We have changed "However, Garimella et al. (2017) noted that the factor was 4 using the Spectrometer for Ice Nuclei, and it varied in the range from about 1 to 10, which indicates that this is one of the major problems in quantifying the formation of ice in numerical models." to
> "However, Garimella et al. (2017) noted that the calibration varied from 1.5 to 9.5 because of the lower relative humidity with respect to water than the intended values if aerosol deviated from the laminar flow, which indicates that this is one of the major problems in quantifying the formation of ice in numerical models."

Line 180: 'DeMott et al. (2010) based on all available data' . This statement is wrong. This

parameterization takes into account several datasets from different regions, but definitely not all available INP data.

> **Reply:** It has been changed to "DeMott et al. (2010) based on several datasets from different regions"

Line 186: Explain PAUKERTD set-up. How ice concentration is enhanced in the dust layer? (actually all tests should be described in detail).

> **Reply:** Please see the reply to the Main point 1.

Line 194-197: Please add a figure that shows the profiles used to initiate the simulations. Documenting the initial thermodynamic conditions is important for comparison with other modeling studies or for other modellers to simulate the same case

> **Reply:** It has been added as Figure 4 in the revised version.

Line 198: 'where another bubble was added 20 min from the starting time'. I am familiar with the fact that a bubble is needed to initiate instability in cloud resolving simulations, but after this I thought that the models can develop turbulence and simulate multi thermals within a 15x 15 km region. What is the physical explanation for the second bubble? What are the model limitations in representing boundary-layer dynamics? Does grid spacing has any impact on this? Also is choice of injection time random or does it have any physical basis (e.g. updraft lifetime). And do results show sensitivity to this choice?

> **Reply:** Please see the reply to the Main point 1 for most of the comment. We did not test the impact of the grid spacing. We think the spacing does not make big difference because decreasing the spacing will represent more smaller eddies but not the thermals from the sub-cloud layer.
>
> We add the following in the text.
> "The injection time of 20 min was chosen when the updraught was about to decay (Figure 11 ). An earlier injection time (e.g., 10 min) of the second bubble only slightly increased the first main updraught and did not change the result significantly."

Line 216: 'The cloud top continued to descend was about 8.5 km (T ~ - 18 C) by 40 min' Please rephrase, it does not make any sense

> **Reply:** It has been changed to "'The cloud top continued to descend to about 8.5 km (T ~ - 18 °C) by 40 min"

Line 217: you refer to the ice crystal concentrations? (graupel is also ice)

> **Reply:** It has been changed to  "The ice crystal concentration".

Line 221: please fill the empty brackets

> **Reply:** The brackets have been deleted.

Line 251: What do you mean 'without the dust layer being considered'? Is the parameterization applied only up to a certain level?

> **Reply:** It has been changed to "when the Bigg (1953) scheme was replaced by the Paukert and Hoose (2014) scheme (*Paukert*)." Some deatails have been added at the end of Section 3.2, i.e., Experimental design, "To investigate the effect of the dust as INP, the Bigg (1953) scheme was replaced by the Paukert and Hoose scheme (2014) since  the Bigg scheme is for general INP types, but the Paukert and Hoose scheme considers different INP types. The *Paukert* run used the mineral dust parameters in the Paukert and Hoose scheme. The *Paukert-dust* run was same as the *Paukert* run except

that the INP numbers were increase by a factor 3.3 in the layer between 2 – 3 km where the dust layer was presented (Figure 1e).”

Line 261: replace 'next' and 'later' with 'in Section…'
**Reply:** It has been changed to “in Sections 4.3 – 4.5.

Line 263: ' due to different microphysical schemes' , you use the same microphysical scheme (Morrison). Please rephrase
**Reply:** It has been changed to “To investigate the causes of the change in secondary ice production sensitivity tests were used  to introduce variations to onset freezing temperature and freezing efficiency”

Line 295: maybe specify that you refer to the ice crystal concentration
**Reply:** It has been changed to “the ice crystal number concentration”.

Line 306: 'enhancing the efficiency' of what?
**Reply:** It has been changed to “enhancing the freezing efficiency”.

Line 314: 'increased approximately above 8 km and decrease' , ' There was less riming', sometimes past tense is used to describe results, and some other times present tense. Please be consistent throughout the whole manuscript
**Reply:** We have checked the text and used the past tense for the simulation throughout the text.

Line 319-320: further clarifications on the implemented modifications are needed
**Reply:** “To account for the dust layer between 2 and 3 km (Figure 1), we modified the INP number to drop number ratio in the Paukert scheme.” has been changed to “To account for the dust layer between 2 and 3 km (Figure 1e), we modified the INP number to drop number ratio by a factor of 3.3 in the Paukert scheme.” Please see the reply to the Main point 1 for more explanation.

Line 324: 'the different microphysical schemes affect ', you use Morrison scheme in all runs.
**Reply:** It has been changed to “The above results indicate that the freezing rate and onset temperature affect the secondary ice production. However, none of them...”

Line 326: ' It would be incorrect not ...'. Remove this sentence or rephrase (e.g. 'it is important to consider... since both play a critical role in ice production', sth like this)
**Reply:** It has been changed to “It is important to consider both the dynamics and microphysics and their interactions since both play a critical role in ice production”.

Line 328: maximum values of what parameter
**Reply:** It has been changed to “maximum values of the vertical velocity, the raindrop concentration, the graupel particle concentration, and the ice crystal number concentration.”

Line 335: ...lasted for a longer time, approximately 30 min (rephrase)
**Reply:** It has been changed to “ lasted for approximately 10 min longer than the first updraft”.

Line 340: 'but there left some' , please rephrase
**Reply:** It has been changed to “ there were few raindrops remaining near cloud top”.

Lines 259-360: ' in the zone to 121 L-1', please rephrase

**Reply:** It has been changed to "one maximum being 84.9 L$^{-1}$ at 58 min and z = 6.9 km and the other maximum being 121 L$^{-1}$ at 69 min and z = 6.75 km."

Lines 363-371: I suggest to add a third panel with temperature timeseries in Figure 12 and include it in the discussion here.

**Reply:** We have not included the temperature time series because the probe became wet in cloud and the values cannot be used

Line 384: 'The shattering mechanism may be most efficient between -10 C and -15 C', yes but ice fragments generated by this mechanism at higher altitudes can be transferred at the lower cloud levels, examined here, right?

**Reply:** "The shattering mechanism may be most efficient between -10 °C and -15 °C." has been changed to "The shattering mechanism may be most efficient between -10 °C and -15 °C, and ice fragments generated by shattering may be transferred to lower or higher altitudes due to the updraughts and downdraughts. "

Line 385: 'Although the Knight mechanism operates at the similar temperature range, there is no parameterization of the process' Why the Phillips et al. (2017) parameterization does not account for this mechanism? They have developed a scheme that takes into account different ice types and habits. The one referred as 'planar' ice category in their study includes needles.

**Reply:** " Although the Knight mechanism operates at the similar temperature range, there is no parameterization of the process." has been changed to "The Knight mechanism operates at the similar temperature range, and future studies can investigate the relative importance of this mechanism using new parameterisations accounting for the ice–ice collision processes (e.g. Phillips et al., 2017)"

Line 368-369: 'Fragments of frozen drops were found, but not in a great amount (figures not shown)'. This is important evidence and should be mentioned above, where the possible contribution from other SIP mechanisms is discussed. You cannot assess the contribution of the drop-shattering process only from images, so your conclusion that this mechanism is insignificant is not necessarily accurate.

**Reply:** The sentence has been modified to "Fragments of frozen drops were found, but not in a great amount, compared with the amount of columnar crystals, although the exact number concentration needed to be determined."

We also changed " it is likely that multi-thermals play an important role in producing very high concentrations of secondary ice particles in some tropical clouds", in the last sentence of the abstract, to "It is possible of course that several mechanisms, some of them only recently being discovered, may be responsible for producing the ice particles in clouds. This study highlights the fact that the dynamics of the clouds likely play an important role in producing high concentrations of secondary ice particles in clouds."

Lines 370-371: 'However, there is no causal evidence and we cannot rule out other mechanisms of secondary ice production'. Maybe refer to Qu et al. (2020) who showed that several SIP mechanisms can operate within a convective cloud.

**Reply:** The paper by Qu et al (2020) has been cited. We have changed the sentence "However, there is no causal evidence and we cannot rule out other mechanisms of

secondary ice production." to "However, there is no causal evidence of other mechanisms of secondary ice production. Qu et al. (2020), for example, showed that several SIP mechanisms can operate within a convective cloud."

Line 394: 'The Morrison microphysics scheme was applied scheme was applied to all runs. Maybe just clarify 'the defaultfor the control runMorrison scheme'
        **Reply:** It has been changed.

Technical corrections:
Line 117: involved an (not in)
Line 124: to measure (not measures)
Line 125: remove 'so that'
Line 128: to measure instead of to measurement
Lines 134-141: it is Figure 1 you are referring to, not Figure 2
Line 139: I don't understand to which panel of Figure 1 you are referring to
Line 287: correct 'increasei7'
Line 312: PAUKERT and CTL
Line 321: to produce secondary ice concentrations similar...
Line 325: observations
Line 344: concentrations
Line 406: 'resulted in even lower' instead of 'had'

        **Reply:** All the above technical corrections have been made.

References:
Qu, Y., Khain, A., Phillips, V., Ilotoviz, E., Shpund, J., Patade, S., and Chen, B.: The role of ice splintering on microphysics of deep convective clouds forming under different aerosol conditions: Simulations using the model with spectral bin microphysics. J. Geophys. Res. Atmos., 125, e2019JD031312. https://doi.org/10.1029/2019JD031312, 2020

Lawson, R. P., Woods, S., and Morrison, H.: The Microphysics of Ice and Precipitation Development in Tropical Cumulus Clouds, J. Atmos. Sci., 72, 2429–2445, https://doi.org/10.1175/JAS-D-14-0274.1, 2015

Lawson, P., Gurganus, C., Woods, S., and Bruintjes, R.: Aircraft Observations of Cumulus Microphysics Ranging from the Tropics to Midlatitudes: Implications for a "New" Secondary Ice Process, J. Atmos. Sci., 74, 2899–2920, https://doi.org/10.1175/JAS-D-17-0033.1, 2017.

Heiblum, R. H., et al. (2016), Characterization of cumulus cloud fields using trajectories in the center of gravity versus water mass phase space: 1. Cloud tracking and phase space description, *J. Geophys. Res. Atmos.*, 121, 6336– 6355, doi:10.1002/2015JD024186.

Heus, T., Jonker, H. J. J., den Akker, H. E. A. V., Griffith, E. J., Koutek, M., and Post, F. H.: A statistical approach to the life cycle analysis of cumulus clouds selected in a virtual reality environment, J. Geophys. Res.-Atmos., 114, D06208, doi:10.1029/2008JD010917, 2009.

Blyth, A. M. and Latham, J.: A multi-thermal model of cumulus glaciation via the Hallett-Mossop process, Q. J. Roy. Meteorol. Soc., 123, 1185–1198, 1997.

CM1 webpage: https://www2.mmm.ucar.edu/people/bryan/cm1/.

---

## Author Comment (AC2)

Response

We thank the reviewer for reviewing our manuscript and the helpful and constructive suggestions. We have considered the comments carefully. Please find below a detailed point-by-point response to all comments.

**Specific comments:**

1. To increase the readability of the manuscript, I suggest renaming the simulations and use some meaningful names. For example: 'control', 'double-HM', 'early onset1', 'cooper10x', 'early onset1 & cooper10x', 'early onset1 & 100xINP', 'early onset2 & 100xINP', 'Demott', 'early Demott ', 'Demott 10xINP, 'multi-thermals' etc. It's hard to follow the results and conclusions with current names.
   **Reply:** Those have been changed throughout the manuscript.

2. One of the major objectives of the paper is to investigate whether multiple thermals in the clouds could explain the observed ice concentration. Therefore, some discussion about previous studies on the conditions favorable for secondary ice production is expected (e.g. graupel fall velocity, updraft speed, broader drop size distribution, etc). (Reference: Cloud Conditions Favoring Secondary Ice Particle Production in Tropical Maritime Convection, Andrew Heymsfield1, and Paul Willis). What are the limitations of the current understanding of these processes? How your study is going to explore these uncertainties.
   **Reply:** We agree with the reviewer's point and have added the following in the paragraph on thermals in the Introduction.
   "The operation of the HM process needs the coexistence of graupel, large drops (> 25 µm) and small drops (< 12 µm) in the temperature zone of -3 °C – -8 °C. Previous studies have investigated the conditions favourable for secondary ice production: moderate vertical velocities to allow graupel particles falling into the HM zone, availability of both large and small drops for riming (e.g., Huang et al., 2008; Heymsfield and Willis, 2014; Huang et al., 2017). It is important to consider the possibility that multiple thermals will ascend through the HM zone because of the additional source of cloud drops that can rime onto graupel particles. "

3. The default parameterization for primary ice nucleation used in the Morrison scheme is based on Cooper, 1986. The authors need to discuss the ice nucleation modes, supersaturation ranges w.r.t. water and ice for this scheme. It should be clear that which ice nucleation modes are active with the primary ice nucleation. Whether you have changed the onset temperature for only immersion-freezing mode?
   **Reply:** The following has been added: "The Morrison scheme has several ice freezing modes, including immersion freezing, deposition freezing as a function of supersaturation with respect to water and ice for this scheme, contact freezing,

homogeneous freezing, and the secondary ice production by the HM process. For relaxation and enhancement sensitivity simulations, we only modified the immersion freezing mode."

4. It will be good to have a separate section for the ICE-D Observation (Section 2). The text after line number 91 to 106 from the introduction and current section 2 can be merged in the new section. Include the brief discussion on quality control of ice number concentration from 2DC-data. What is the size range of ice particles from observations mentioned in the paper? Whether the particles smaller than 200 µm were considered? Show that the conditions were favorable in the observed clouds for secondary ice formation. e.g. presence of drop larger than 24 µm size, presence of graupel particles in HM zone, etc in the same section. Figures 12 and 13 and relevant results can be added to this section. It should be clear that the comparison between model-simulated ice particles and observations is for particles of a similar size range.

> **Reply:** We thank the reviewer for this suggestion and have thought about it carefully. The paragraph (Lines 92-100 in the original version) is about the campaigns in this region and the difference between the ICE-D and the previous campaigns. The next paragraph highlights the high concentration of ice particles. Those paragraphs provide the aims of this study, which is an essential element of the introduction. It will leave a gap in the introduction if the paragraphs move to the next section. For this reason, we think it is better for them to remain in their current places. But, "A suite of instruments on board the UK research aircraft FAAM (Facility for Airborne Atmospheric Measurements) BAe 146 measured the information about cloud microphysics, aerosol particles, and other atmospheric variables (see Price et al., 2018, Liu et al., 2018, and Lloyd et al., 2020 for further descriptions)." has been moved into Section 2.
>
> Figures 12 and 13 are in the discussion section together with the possible other mechanisms. Besides, the text length is in rough balance in this way. They are better in their current places.

5. Line 88: Authors mentioned previous work by Blyth and Latham (1997) on the role of multi-thermal in HM process. Mention in which aspects your simulations are different than theirs. What are the novel approaches of your study? Does your study agree/disagree with their simulations? How realistic are the simulated thermal in the model? The authors need to add some discussion on this.

> **Reply:** The following has been added at the end of Section 4.5: "The results are consistent with the findings of Blyth and Latham (1997) in that multi-thermals can significantly enhance the secondary ice production. A conceptual representation of the kinematics was used in the detailed microphysics model described by Blyth and Latham (1997), whilst the present study employed a three-dimensional cloud model with detailed cloud microphysical processes. There have been a few studies of

thermals in shallow convective clouds (e.g., Heus et al., 2009; Heiblum et al., 2016). It is impossible to make a direct comparison of thermals between the deep convective cloud in this paper and those shallow clouds, but similar features were found, such as enhanced vertical velocities and cloud mass associated with the thermals."

6. In the conclusions, the authors mentioned that the multiple thermal still cannot reproduce the observed highest concentration of ice particles. There should be more discussion about it.

Other secondary ice mechanisms may not be fully absent at temperatures warmer than -10$^o$ Observations indicated the presence of fragments of frozen drops. The parametrization of secondary ice particles from frozen drops by Phillips et al. 2017 indicates that at those warmer temperatures this process might have a considerable contribution to the secondary ice formation.

> **Reply:** The following has been added in the text: "Recent development of the parametrization of secondary ice particles from frozen drops by Phillips et al. (2017) indicates that this process might have a considerable contribution to the secondary ice formation at temperatures greater than -10 °C. Future research will undoubtedly include this parametrization. It should also be noted that research continues on mechanisms that can cause an enhancement of ice particles (e.g. James et al., 2021)." "

**Minor/technical comments:**

**Line 76:** Cite recent review articles on secondary ice formation e.g. Review of experimental studies of secondary ice production: A Korolev, T Leisner Atmospheric Chemistry and Physics 20 (20), 11767-11797; A new look at the environmental conditions favorable to secondary ice production: A Korolev, I Heckman, M Wolde, AS Ackerman, AM Fridlind, LA Ladino, Atmospheric Chemistry and Physics 20 (3), 1391-1429.

> **Reply:** Those papers have been added in the text.

**Line 91:** Define ICE-D in the text also even if it is defined in the abstract.

> **Reply:** It has been added in the text.

**Line 94:** Rephrase/remove the sentence 'However….

> **Reply:** As suggested by another reviewer, this sentence has been deleted.

**Line 101:** Is FAAM defined earlier?

> **Reply:** It has been added here.

**Line 119**: Full stop is missing after … and aerosol

> **Reply:** It has been added.

**Line 134:** Correct the figure number. It is Figure 1a. Also add the axes titles (Lat, Long) on the plot. Correct the figure numbers in the rest part of the paragraph e.g. line number 135, 138, 141.

> **Reply:** The figure numbers have been corrected, and the axes titles have been added.

**Figure 3:** Check the legends for 2DC data. Add the axes title in X-axes (Ice particle number concentration). Mention the size range of ice particles considered here.

> **Reply:** It has been changed The caption has been changed to "Figure 3, Time series of the concentration of ice particles ($L^{-1}$) in the size range of 50 -1280 µm measured with 2D-S Stereo *Probe* and vertical velocity ($ms^{-1}$) between 15:48:05 UTC and 15:49:30 UTC on 21 August 2015. "

**Figure 4:** Add axes titles.

> Reply: The axis titles have been added..Please note that a new figure has been added before this figure which is now Figure 5 in the revised version.

**Line 150:** Mention the lowest diameter of ice particles considered in estimating their concentrations. Are you considering particles smaller than 200 um from the observations?

> **Reply:** The concentration of ice particles in the figure only include those with irregular shapes (non-spherical) measured with the 2DS. The sentence has been changed to "The maximum concentrations of ice particles (i.e., non-spherical in shape) in the size range of 50 -1280 µm were ..."

**Line 151:** Change -6,8 $^{o}C$ to -6.8 $^{o}C$.

> **Reply:** It has been changed.

**Line 163:** Add reference for Hallet Mossop process

> **Reply:** The original paper of Hallett and Mossop (1974) has been added. Also added is the paper by Cotton et al. (1986) in which the treatment of the HM process has been used.by CM1.

**Line 209**: change 14C to -14$^{0}C$

> **Reply:** It has been changed.

**Line 213:** Change the graupe to the graupel

> **Reply:** It has been changed.

**Line 221:** Check an empty bracket after velocity

> **Reply:** It has been deleted.

**Line 254:** Do you mean 'starting freezing temperature'.

> **Reply:** It has been corrected.

**Line 255:** How do you know the ice particles observed during aircraft observations are only originating from secondary ice processes.

**Reply:** It has been changed to "the primary and the secondary ice production".

**Section 4.3:** Be specific about simulations involving changes in freezing efficiency. Mention clearly what are the changes made for this.

**Reply:** We have added the following in the Experimental design section. "The Morrison scheme has several ice freezing modes, including immersion freezing, deposition freezing as a function of supersaturation with respect to water and ice for this scheme, contact freezing, homogeneous freezing, and the secondary ice production by the HM process. For relaxation and enhancement sensitivity simulations, we only modified the immersion freezing mode. The aims of the sensitivity simulations are summarised as follows. *early onset1* examined the effect of active INPs at higher temperatures on secondary ice production when the onset temperature was increased to -3 °C. *Cooper10x* explored the effect of more INPs (i.e., the freezing efficiency was multiplied by 10). *early ohnset1 & Cooper10x* combined effects of the above two, while *early onset1 & 100xINP* and *early onset2 & 100xINP* probed the effect of even higher loadings of INPs. The DeMott scheme (2010) was examined in runs *Demott*, *early Demott*, and *Demott 10xINP*. To investigate the effect of the dust as INP, the Bigg (1953) scheme was replaced by the Paukert and Hoose scheme (2014) since the Bigg scheme is for general INP types, but the Paukert and Hoose scheme considers different INP types. The *Paukert* run used the mineral dust parameters in the Paukert and Hoose scheme. The *Paukert-dust* run was same as the *Paukert* run except that the INP numbers were increase by a factor 3.3 in the layer between 2 – 3 km where the dust layer was presented (Figure 1e). Finally, the effect of multi-thermals on secondary ice production was examined in *multi-thermals* when a second bubble of 2 °C was added after 20 min into the simulation."

**Line 266:** Correct 'TLXTEN'

**Reply:** It has been corrected. Please note names of all simulations in Table 1 and in the text in response to the other reviewer.

**Line 394:** I think the Morrison microphysics scheme was applied for all the runs. It was modified for other sensitivity tests. Make this point clear in the text.

**Reply:** "The Morrison microphysics scheme was applied for the control run" has been changed to "The Morrison microphysics scheme was applied for the simulations."

**Line 420:** Add some of the challenges in measuring full spectra of INP and CCN.

**Reply:** It has been added.

**Figure 2:** Check the X-axes title. What is HI? Is it defined in the text? Also, mention the Temperature on the color bar in the box. Also, change the title of X axes to Altitude (above MSL??) (m). Check its position w.r.t plot.

**Reply:** It has been changed. "The colour bar represents temperature." has been added in the caption."

**Figure 7:** Check the spelling of 'ratio' in the figure title.

>          **Reply:** They have been changed.

**Table 1:** RLX3X100 is the experiment where you relax the onset temperature in the **ice nuclei number** concentration and not the ice concentration. Similar to RLX2X100.

>          **Reply:** They have been changed to "the ice nuclei number concentration".

**References**

James, R. L., Phillips, V. T. J., and Connolly, P. J.: Secondary ice production during the break-up of freezing water drops on impact with ice particles, Atmos. Chem. Phys. Discuss. [preprint], https://doi.org/10.5194/acp-2021-557, in review, 2021.

Heymsfield, A. and Willis, P.: Cloud conditions favoring secondary ice particle production in tropical maritime convection, J. Atmos. Sci., 71, 4500–4526, https://doi.org/10.1175/JAS-D-14-0093.1, 2014.

Huang, Y., Blyth, A., Brown, P., Choularton, T., Connolly, P., Gadian, A., Jones, H., Latham, J., Cui, Z., and Carslaw, K.: The development of ice in a cumulus cloud over southwest England, New J. Phys., 10, 105021, https://doi.org/10.1088/1367-2630/10/10/105021, 2008

Huang, Y., Blyth, A. M., Brown, P. R. A., Choularton, T. W., and Cui, Z.: Factors controlling secondary ice production in cumulus clouds, Q. J. Roy. Meteor. Soc., 143, 1021–1031, 2017.

---

## Author Comment (AC3)

Replay to Reviewer 3.

The authors discussed observations of ice crystal concentrations measured in-situ during the ICE-D field campaign in Cape Verde during August 2015. It was suggested that, based on INP measurements, primary nucleation alone could not explain the high ice crystal concentrations and so secondary ice processes, e.g. rime splintering, occurred in the convective cloud.

Simulations with one or multiple thermals were performed with several configurations of the heterogeneous ice nucleation scheme. The maximum ice crystal concentrations were than compared with the ones measured during aircraft passes through the cloud.

**Major Comments**

**1.** The vertical and horizontal extent as well as the lifetime of the modeled cloud are not compared to the observed cloud, for example by utilizing weather radar data.

> **Reply:** A ground-based mobile radar was deployed, but the clouds were 150 – 200 km from the radar and so due to the beam width and curvature of the Earth, it was not possible to perform detailed comparisons. However, the cloud penetrations were made following a ascending cloud top in order to detect the first ice and in the HM zone to measure the secondary ice production. The top of the simulated clouds was close to the observed clouds. Therefore, we could not directly compare the cloud extents.

**2.** The microphysics of the model are also very sensitive to the initial conditions, especially the vertical profile of water vapor. The authors used initial profiles combined from dropsonde measurements and NCEP/NCAR reanalysis data, but did not show these profile or discuss which parts of the simulated clouds are located in the region approximated with reanalysis data. The authors should put more emphasis on linking observations with model results and discussing uncertainties of initial conditions.

> **Reply:** The initial profile of the potential temperature and the water mixing ratio has been added. In the discussion section, we added the following in the text. "A simulated cloud with a model is sensitive to its initial conditions. A perfect initialisation requires to follow the trajectory of the cloud in time and space to get the vertical variation of thermodynamic variables before its formation. The initial conditions we could get in close proximity to reality was from the aircraft profile run after it took off to reach the clouds. The initialisation based on the combination of aircraft measurement and reanalysis was a source of uncertainty. However, the major conclusion of this study is that the multi thermal is the only way to get enough ice. The initial conditions only have to be roughly correct to get a cloud that goes to correct height and has the same magnitude of updraft velocities and horizontal extent."

**3.** A major part of the focuses on evaluating the contribution of secondary ice from rime splintering. The authors identified secondary ice by observing the temperature zone where the Hallett-Mossop (HM) processes is active, e.g. between -3 and -8 °C., and attributed ice crystals in that temperature range to rime splintering. However, transport processes, e.g. advection and sedimentation of ice crystals from the colder part of the cloud into the HM zone, are not discussed.

> **Reply:** As shown in Figure 5d, the high concentrations of ice crystals are in two regions. One is between 27 – 34 min and 6 – 7 km, and the other is above ~ 8 km outside the HM zone.

Figure 5c shows that graupel particles fall into the same region as that in Figure 5d of the high ice concentration in the HM zone. The high ice concentration is indeed by the HM process. The reply to the next point also reinforces this point.

**4,** Furthermore a majority of the sensitivity experiments increased the onset freezing temperature from -8 °C to – 3 °C or higher, hence heterogeneous nucleation is now possible inside the HM zone. This makes it very difficult to distinguish between ice from rime splintering and primary nucleation in that temperature range. The authors should build a stronger argument that increased ice crystal concentrations in the HM zone did indeed originate from enhanced rime splintering. For example sensitivity simulations without rime splintering could be performed and included in the comparisons.

> **Reply:** We have conducted an additional simulation without the rime splintering process. The figure  below shows the time-height variation of the ice crystal number concentration in (a) the *early onset1* run in which the onset freezing temperature being -3 °C, rather than -8  °C, (b)  *the early onset1* & noHM run in which the onset freezing temperature being -3 °C, rather than -8  °C, but the HM process was switched off, and (c) the difference between the simulation with the HM process and the simulation without the HM process. The increase in ice number concentration is clearly seen between 27 – 34 min and 6 – 7 km in (a) and (c), but not in (b). This clearly indicates that the increased ice crystal concentrations in the HM zone did indeed originate from enhanced rime splintering. We also have added the simulation in Figure 6.

[Figure]

> **Reply:** It has been changed to "The INP concentrations are highly variable at a temperature depending on aerosol properties, dynamic and humidity conditions."

**Line 76:** there are also newer review papers relevant for secondary ice processes e.g. Korolvel et al. (2020)

> **Reply:** Two papers have been added here. It has been changed to "see Field et al. (2017), Korolev and Leisner (2020), and Korolev et al. (2020). "

**Line 93:** missing full stop after Field et Al

> **Reply:** The sentences have been deleted as suggested by another reviewer.

**Line 119:** missing full stop after aerosol

> **Reply:** It has been added.

**Line 134-143:** there are a few mismatches in the figure references

> **Reply:** All mismatches have been corrected.

**Line 151-153:** Fig. 3 does not provide information about temperature. The statement that ice appeared only at T < -10 °C is supported with Fig. 2.

> **Reply:** It has been changed. The following has been added: "The observation indicates that the ice concentrations were a few tens per litre at derived temperatures in cloud between 0 and −2 ˚C (Lloyd et al., 2020). Therefore, secondary ice production most likely occurred"

**Line 153-154:** this section has to be expanded to support the statement that measured ice concentrations were lower than what primary nucleation would suggest since Fig. 2 and 3 do not show INP concentrations

> **Reply:** It has been changed to "much higher than the predicted by most primary ice production parameterisation schames."

**Line 164:** other relevant microphysical processes, which do not require interaction of cloud particles, like sedimentation and particle growth by deposition of water vapor should be mentioned. Also the Hallett-Mossop temperature zone is mentioned multiple times in this work and the temperature range should be explicitly stated in this section

> **Reply:** The processes of sedimentation and particle growth by deposition of water vapour have been added. The temperature range of the HM process has been added too. The text has been changed to "The microphysical processes include drop activation, condensation, evaporation, collision and coalescence, sedimentation of cloud particles, particle growth by deposition of water vapour, primary freezing modes of deposition/condensation, contact, and immersion, and secondary freezing through the riming-splintering (the Hallett-Mossop) process (Hallett and Mossop, 1974; Cotton et al., 1986), in the temperature range of -3 to -8 °C with a maximum at -5 °C, and the transition and interaction between different species."

**Line 191-193:** include information about model time steps and intervals of writing output. The latter is important for how clearly you can identify nucleation events in your data.

> **Reply:** The following has been added: "The time-step is 2 sec, and the output frequency is 1 min."

**Line 209:** -14 °C instead of -14C

> **Reply:** It has been changed.

**Line 213**: misspelled graupel

> **Reply:** It has been changed.

**Line 216:** check grammar of last sentence in this line

> **Reply:** It has been corrected.

**Line 221:** parenthesis without content

>**Reply:** It has been deleted.

**Line 266:** this statement should be better connected to the figure, that actually shows the increase in ice crystal concentrations

>**Reply:** The sentences from the beginning of the paragraph have been changed to "Sensitivity tests were used to investigate the effect of varying the onset freezing temperature and freezing efficiency on secondary ice production. The differences of several microphysical properties between a sensitivity and the control simulations are examined. There was a significantly higher concentration of secondary ice particles produced in *early onset1 & Cooper10x* compared to the *control* run. Figure 7 shows the detailed differences between the two runs."

**Line 288:** misspelled increase

>**Reply:** It has been changed.

**Line 332:** remove 'also' before 'approximately'

>**Reply:** It has been deleted.

**Line 334:** remove space between m and s$^{-1}$

>Reply: It has been deleted.

**Line 340:** check grammar in second part of the sentence

>**Reply:** It has been changed to " Some raindrops were transferred to graupel particles via direct freezing, but there were few raindrops remaining near cloud top."

**Line 367:** panel d in Fig. 12 does not exist

>**Reply:** It has been corrected.

**Line 367-377:** there are no ice particles shown in Fig. 13

>**Reply:** The right panel shows images of ice particles, such as columns and small non-spherical particles.

**Fig 4.:** labels of isotherms are barely visible, but important for interpreting the plot. Purple is misspelled in the plot label.

>**Reply:** Please note it is Fig. 5 in the revised version since a new figure is added. It has been changed with thick lines of temperature. Purpple has been changed to purple. Figure 12 in the revised version has been changed in the same way.

**Fig. 5:** color bar should be changed so that the colors below the dark blue are more continuous. Also a logarithmic plot would be good since ice crystal concentrations below 1 /L are hidden, but important in how ice in the top of the cloud and the HM zone connect.

**Reply:** Please note it is Fig. 6 in the revised version since a new figure is added. It has been changed. Two more intervals, 0.5 L$^{-1}$ and 0.1 L$^{-1}$ have been added so that it shows how ice in the top of the cloud and the HM zone connect.

**Fig. 7:** RLX10 is called RLXTEN in the text

**Reply:** Please note it is Fig. 8 in the revised version since a new figure is added. It has been changed. Another reviewer suggested rename the control and the sensitivity simulations to increase the readability. We agree with the reviewer and have changed the names throughout the text, the table, and the figures. Please see the table for new names.

**Fig. 8:** the color bar is very hard to read. A continuous color bar should be used instead. Also the plot label misses the information that maximum ice crystals concentrations are shown. RLX10 is called RLXTEN in the text

**Reply:** Please note it is Fig. 9 in the revised version since a new figure is added. The colour bar has been changed. The figure shows the DIFFERENCE between in number concentration. The x-axis and y-axis are time and altitude, respectively, as shown in the replotted figure. RLX10 and RLXTEN have been changed to a new name. Please see the above reply.

**Fig. 9:** color bar should be changed so that the colors below the dark blue are more continuous.

**Reply:** Please note it is Fig. 10 in the revised version since a new figure is added. It has been changed.

**References**

Korolev, A. and Leisner, T.: Review of experimental studies of secondary ice production, Atmos. Chem. Phys., 20, 11767–11797, https://doi.org/10.5194/acp-20-11767-2020, 2020.

---

## Referee Report (RR1)

REVIEW of the article "**Multi-thermals and high concentrations of secondary ice: A modelling study of convective clouds during the ICE-D campaign**"

by Cui et al.

The manuscript is improved in many aspects as compared to the previous version. However, there are still some grammatical and other minor mistakes that authors need to address before accepting it for final publication.

**Minor comments:**

Abstract: line 15- change parametrization to parameterization

Line 24: change ice nucleating to ice-nucleating

Line 30: delete 'being'

Line 46: change minerology to mineralogy

Line 49: **the** ice-nucleating sites…

Line 50: **the** formation ….

Line 53: change 'do not active' to 'do not act'

Line 58: feldspar **are** particularly

Line 70: **the** mechanical breakup

Line 80: delete the from ' the future research'

Line 97: change falling into to fall into

Line 105: change large scale to large-scale

Line 107: change 'projects of layer..' to project on layer….

Line 112: remove 'as' before greater.

Line 113: You have used secondly two times. Modify the sentences.

Line 121: is i**ncluded** in section 3 or **given** in section 3

Line 124: add space between UK's and FAAM

Line 126: change on board to onboard

Line 129: remove the full stop after 'micrometers'

Line 130: define 2D-S

Line 133: polarisation

Line 146: define MSL

Line 156: hundred **meters**

Line 159: change 'of several passes from the aircraft measurements' to in several passes based on the aircraft measurements.

Line 163-164: This is not very clear. What is the reason behind 'secondary ice production most likely occurred'.

Line 172-175: Split this into separate sentences.

Line 175: is '.and the transition and interaction between different species' is part of the previous statement.

Line 177: One of the objectives **of our study** is to. Remove 'the' after freezing efficiency on

Line 178: As **mentioned** in the introduction ..

Line 180: Similarly to what? This statement has no connection with the previous one.

Line 181: … field measurements of **INPs…….** Replace 'the influence on cloud simulations' by 'its influence on the simulated cloud forcing in a global model.'

Line 189: Replace ' As a result of this uncertainty by 'Considering these uncertainties'

Line 194: replace model runs by tests

Line 194-196: This statement is not very clear. Please split into separate sentences

Line 198: **The** early onset examined

Line 199: The Cooper10x ….

Line 200: The early onset 1 &…

Line 203: Which general type of INPs?

Line 205: the INP numbers were **increased.** Why a factor of 3.3, Any particular reason?

Line 206: 2-3 km

Line 212: was 1 **minute.**

Line 215: water **vapor** mixing ratio

Line 225: a rate **of** about 150

Line 228-229: Replace 'At 25 min, the cloud top reached about 8 km' by 'The cloud top had reached around 8 kilometers at 25 minutes'.

Line 232: Delete 'down'

Line 246: .. with a maximum **of**

Line 251: change wasidentical to was identical. That of **the** control run…

Line 259: high variability in **the** INP population

Figure 3: IN captions replace the comma with a full stop after 'Figure 3'.

Figure 4: In captions change ration to ratio. Also, follow the same structure of caption throughout the manuscript. e.g. (a) potential temperature and (b) mixing ratio

Figure 13, 14: In the caption, these figures are mentioned as Figur N13, N14. In Figure 14, please mention the width of each CIP images strip. It will give an idea about the particle size.

Figure 4: Altitude is above mean sea level/above ground?

Line 263: Move 'as plotted in Figure 7' at the end of the statement. Remove comma after the freezing efficiency. Change wass to was.

Line 278: replace 'similar concentrations of the primary and the secondary ice production to the observations made in some passes' by 'total ice concentration observed in some passes'

Line 287: Replace 'a sensitivity and the control' by 'the sensitivity tests and the control'

Line 289: full stop after 'between the two runs'

Line 291: replace 'and the maximum was' by 'reaching a maximum concentration of about'

Line 295: Don you mean cloud **water** mixing ratio. Please clarify

Line 304-306. Please clarify this statement. It is not clear what the authors want to say here.

Line 310: change mixing ration to mixing ratio

Line 323: further **increased**

Line 324-325: Rewrite the statement. more/less can be written as more (less). Replace higher temperature with higher temperatures.

Line 327: there **was a** slight increase at the upper levels

Line 332: secondary ice in **the** concentration

Line 344: by modified you mean by increasing. Please clarify. Why 3.3?

Line 346: that dust alone **as a source of INPs** was

Line 349: freezing rate and onset temperature of what?

Line 350: delete the from 'impact of the cloud'

Line 361: change updraght to updraft.

Line 383-384: delete 'down'

Line 422: Need a full stop after updraughts and downdraughts. What is the Knight mechanism?

Line 423: delete the before 'similar temperature range'

Line 427: **The** recent development ……

Line 442: **the** mountainous **regions**

Line 447: replace 'to follow' by following

Line 452: goes to **the** correct height

Line 465: remove 'the' before 'observed'.

Line 473: at **a** higher altitude

Line 487: delete 'the' before 'the cloud microphysics'. Remove full stop before (Field et al..)

---

## Author Response (AR2)

Responses to Report 3

We thank the reviewer for taking the time to catch the grammatical and minor errors.

**The manuscript is improved in many aspects as compared to the previous version. However, there are still some grammatical and other minor mistakes that authors need to address before accepting it for final publication.**

**Line 15: change parametrization to parameterization**
'parametrization' has been changed to 'parameterization'.

**Line 24: change ice nucleating to ice-nucleating**
'ice nucleating' has been changed to 'ice-nucleating'.

**Line 30: delete 'being'**
'being' has been deleted.

**Line 46: change minerology to mineralogy**
'minerology' has been changed to 'mineralogy'.

**Line 49: the ice-nucleating sites…**
It has been changed to 'the ice-nucleating sites'

**Line 50: the formation .…**
'the' has been added.

**Line 53: change 'do not active' to 'do not act'**
'do not active' has been changed to 'do not act'

**Line 58: feldspar are particularly**
'is' has been changed to 'are'.

**Line 70: the mechanical breakup**
'the' has been added.

**Line 80: delete the from ' the future research'**
'the' has been deleted.

**Line 97: change falling into to fall into**
'falling into' has been changed to 'fall into'.

**Line 105: change large scale to large-scale**
'large scale' has been changed to 'large-scale'.

**Line 107: change 'projects of layer..' to project on layer.…**
'projects of layer' has been changed to 'project on layer'.

**Line 112: remove 'as' before greater.**
'as' has been removed.

**Line 113: You have used secondly two times. Modify the sentences.**
The first sentence has been removed.

**Line 121: is included in section 3 or given in section 3**
    It has been changed to 'given in section 3'.

**Line 124: add space between UK's and FAAM**
    A space has been added.

**Line 126: change on board to onboard**
    'on board' has been changed to 'onboard'.

**Line 129: remove the full stop after 'micrometers'**
    The full stop has been deleted.

**Line 130: define 2D-S**
    It has been defined as 'two-dimensional stereo (2D-S) Probe'.

**Line 133: polarisation**
    The Met Office of the UK uses 'relative depolarisation ratio' rather than 'relative polarisation ratio' Please see
https://catalogue.ceda.ac.uk/uuid/3fdd6ccdce5b4ee887536402edfe6835.

**Line 146: define MSL**
    It has been defined: 'mean sea level (MSL)'.

**Line 156: hundred meters**
    It is the British spelling.

**Line 159: change 'of several passes from the aircraft measurements' to in several passes based on the aircraft measurements.**
    'of several passes from the aircraft measurements' has been changed to 'in several passes based on the aircraft measurements.'

**Line 163-164: This is not very clear. What is the reason behind 'secondary ice production most likely occurred'.**

This comment was also made by Reviewer 2.
    'The observation indicates that the ice concentrations were a few tens per litre at derived temperatures in cloud between 0 and −2 ˚C (Lloyd et al., 2020). Therefore, secondary ice production most likely occurred.' has been changed to 'The observation indicates that the ice concentrations were a few tens per litre at derived temperatures in cloud between 0 and −2 ˚C (Lloyd et al., 2020). It is impossible of course to be certain of the origin of ice particles in such clouds. The fact that the passes were made within a few hundred metres of cloud top, the concentrations of ice particles were higher than expected from typical INP measurements for the estimated cloud-top temperatures, and there was no evidence of higher concentrations of ice particles in the downdraughts suggest that secondary ice production most likely occurred..'

**Line 172-175: Split this into separate sentences.**
    The sentences have been changed to
'The microphysical processes of drops include condensation, evaporation, collision and coalescence, sedimentation of cloud particles, particle growth by deposition of water vapour. The processes involved with ice include primary freezing modes of deposition/condensation, contact, and immersion, and secondary freezing through the riming-splintering (the Hallett-Mossop) process

(Hallett and Mossop, 1974; Cotton et al., 1986), in the temperature range of -3 to -8 °C with a maximum at -5 °C, and the transition and interaction between different species.'

**Line 175: is '.and the transition and interaction between different species' is part of the previous statement.**
  '.and ...' has been changed to ', and ...'.

**Line 177: One of the objectives of our study is to. Remove 'the' after freezing efficiency on 'of our study' has been added.**
  'the' has been deleted.

**Line 178: As mentioned in the introduction ..**
  'As is discussed in the introduction' has been replaced by 'As mentioned in the introduction'.

**Line 180: Similarly to what? This statement has no connection with the previous one.**
  'Similarly, the temperature where freezing begins depends on the INP types.' has been changed to 'The temperature where freezing begins depends on the INP types.'

**Line 181: ... field measurements of INPs....... Replace 'the influence on cloud simulations' by ' its influence on the simulated cloud forcing in a global model.'**
  'the influence on cloud simulations' has been replaced by 'its influence on the simulated cloud forcing in a global model.'

**Line 189: Replace ' As a result of this uncertainty by 'Considering these uncertainties'**
  'As a result of this uncertainty' has been replaced by 'Considering these uncertainties'

**Line 194: replace model runs by tests**
  'model runs' has been replaced by 'tests'.

**Line 194-196: This statement is not very clear. Please split into separate sentences**
  'The Morrison scheme has several ice freezing modes, including immersion freezing, deposition freezing as a function of supersaturation with respect to water and ice for this scheme, contact freezing, homogeneous freezing, and the secondary ice production by the HM process.' has been changed to 'The freezing modes of the Morrison scheme include immersion freezing, deposition freezing as a function of supersaturation with respect to water and ice for this scheme, contact freezing, homogeneous freezing, and the secondary ice production by the HM process.'

**Line 198: The early onset examined**
  'The' has been added.

**Line 199: The Cooper10x .…**
  'The' has been added.

**Line 200: The early onset 1 &…**
  'The' has been added.

**Line 203: Which general type of INPs?**
  'the Bigg scheme is for general INP types' has been changed to 'the Bigg scheme does not consider INP types'.

**Line 205: the INP numbers were increased. Why a factor of 3.3, Any particular reason?**

'...by a factor 3.3 in the layer between 2 – 3 km where the dust layer was presented' has been changed to 'by a factor 3.3 in the layer between 2 – 3 km where the coarse mode concentrations in that dust layer were approximately 3.3 times higher than those above the layer.'

**Line 206: 2-3 km**
'2 - 3 km' has been changed to '2-3 km'.

**Line 212: was 1 minute.**
**'1 min' has been changed** to '1 minute'.

**Line 215: water vapor mixing ratio**
'water mixing ratio' has been changed to 'water vapour mixing ratio'.

**Line 225: a rate of about 150**
'of' has been added.

**Line 228-229: Replace 'At 25 min, the cloud top reached about 8 km' by 'The cloud top had reached around 8 kilometers at 25 minutes'.**
'At 25 min, the cloud top reached about 8 km' has been replaced by 'The cloud top had reached around 8 km at 25 minutes'.

**Line 232: Delete 'down'**
'down' has been deleted.

**Line 246: .. with a maximum of**
'with a maximum' has been changed to 'with a maximum of'.

**Line 251: change wasidentical to was identical. That of the control run…**
A space has been added between 'was' and 'identical'.

**Line 259: high variability in the INP population**
'the' has been added.

**Figure 3: IN captions replace the comma with a full stop after 'Figure 3'.**
The comma has been replaced with a full stop.

**Figure 4: In captions change ration to ratio. Also, follow the same structure of caption throughout the manuscript. e.g. (a) potential temperature and (b) mixing ratio**
The caption has been changed to 'Figure 4. Initial profiles of (a) potential temperature and (b) mixing ratio for the model simulations. Altitude is the height above mean sea level.'

**Figure 13, 14: In the caption, these figures are mentioned as Figur N13, N14. In Figure 14, please mention the width of each CIP images strip. It will give an idea about the particle size.**
N13 and N14 have been changed to 13 and 14, respectively.
'The CIP image width is 960 µm.' has been added in the caption.

**Figure 4: Altitude is above mean sea level/above ground?**
'Altitude is the height above mean sea level' has been added in the caption.

**Line 263: Move 'as plotted in Figure 7' at the end of the statement. Remove comma after the freezing efficiency. Change wass to was.**
'as plotted in Figure 7' has been moved at the end of the statement. The comma after the

freezing efficiency has been deleted. 'wass' has been changed to 'was'.

**Line 278: replace 'similar concentrations of the primary and the secondary ice production to the observations made in some passes' by 'total ice concentration observed in some passes'**

'similar concentrations of the primary and the secondary ice production to the observations made in some passes' has been replaced by 'total ice concentration observed in some passes'.

**Line 287: Replace 'a sensitivity and the control' by 'the sensitivity tests and the control'**

'a sensitivity and the control' has been replaced by 'the sensitivity tests and the control'.

**Line 289: full stop after 'between the two runs'**

A full stop has been added.

**Line 291: replace 'and the maximum was' by 'reaching a maximum concentration of about'**

'and the maximum was' has been replaced by 'reaching a maximum concentration of about'.

**Line 295: Don you mean cloud water mixing ratio. Please clarify**

'cloud mixing ratio' has been changed to ' cloud water mixing ratio'.

**Line 304-306. Please clarify this statement. It is not clear what the authors want to say here.**

"while it was downwards in the control run the axis in the control run went down after 38 min" has been changed to "However, a decrease in the concentration occurred after about 37 min with the minimum being above 8 km.".

**Line 310: change mixing ration to mixing ratio**

'ration' has been changed to 'ratio'.

**Line 323: further increased**

It has been corrected.

**Line 324-325: Rewrite the statement. more/less can be written as more (less). Replace higher temperature with higher temperatures.**

'There were more/less increases in the lower/upper levels when using the DeMott scheme (Demott) because of the slope of DeMott curve, i.e., more IN at higher temperature.' has been changed to 'There were more (less) increases in the lower (upper) levels when using the DeMott scheme (Demott) because of the slope of the the DeMott curve, i.e., more IN at higher temperatures.'

**Line 327: there was a slight increase at the upper levels**

It has been changed to 'there was a slight increase'.

**Line 332: secondary ice in the concentration**

' secondary ice in concentration of several tens per litre' has been changed to 'secondary ice in concentrations of several tens per litre'.

**Line 344: by modified you mean by increasing. Please clarify. Why 3.3?**

It has been changed to 'we increased the INP number to drop number ratio by a factor of 3.3 in the Paukert scheme'.

**Line 346: that dust alone as a source of INPs was**

'dust alone as a source of INPs was not enough to produce the secondary ice concentrations similar to the observations in this case.' has been changed to 'dust alone as a source of INPs was not enough to produce secondary ice concentrations that were similar to the observations in this case.'

**Line 349: freezing rate and onset temperature of what?**
It has been changed to "the freezing rate and onset temperature in the Morrison scheme affect".

**Line 350: delete the from 'impact of the cloud'**
'the' has been deleted.

**Line 361: change updraght to updraft.**
'updraught' is the British spelling. Authors tend to use it.

**Line 383-384: delete 'down'**
'down' has been deleted.

**Line 422: Need a full stop after updraughts and downdraughts. What is the Knight mechanism?**
A full stop has been added. The Knight mechanism is mentioned in the first sentence of the paragraph. To make it clearer, "The Knight mechanism" has been changed to " The Knight mechanism (Knight, 2012).

**Line 423: delete the before 'similar temperature range'**
'the' has been deleted.

**Line 427: The recent development ...…**
'Recent development' has been changed to 'The recent development'

**Line 442: the mountainous regions**
'mountainous regions' has been changed to 'the mountainous regions'.

**Line 447: replace 'to follow' by following**
'to follow' has been changed to 'following'

**Line 452: goes to the correct height**
'the' has been added.

**Line 465: remove 'the' before 'observed'.**
'the observed' has been changed to 'observed',

**Line 473: at a higher altitude**
It has been changed to 'at a higher altitude'.

**Line 487: delete 'the' before 'the cloud microphysics'. Remove full stop before (Field et al..)**
'the' has been deleted. The full stop has been deleted as well.